# Benchmark single-step ethylene purification from ternary mixtures by a customized fluorinated anion-embedded MOF

Yunjia Jiang,[1,4], Yongqi Hu,[1,4], Binquan Luan,[2], Lingyao Wang,[1], Rajamani Krishna, [3], Haofei Ni,[1], Xin Hu[1] & Yuanbin Zhang [1] ✉

Ethylene ($C_2H_4$) purification from multi-component mixtures by physical adsorption is a great challenge in the chemical industry. Herein, we report a $GeF_6^{2-}$ anion embedded MOF (ZNU-6) with customized pore structure and pore chemistry for benchmark one-step $C_2H_4$ recovery from $C_2H_2$ and $CO_2$. ZNU-6 exhibits significantly high $C_2H_2$ (1.53 mmol/g) and $CO_2$ (1.46 mmol/g) capacity at 0.01 bar. Record high $C_2H_4$ productivity is achieved from $C_2H_2$/ $CO_2$/$C_2H_4$ mixtures in a single adsorption process under various conditions. The separation performance is retained over multiple cycles and under humid conditions. The potential gas binding sites are investigated by density functional theory (DFT) calculations, which suggest that $C_2H_2$ and $CO_2$ are preferably adsorbed in the interlaced narrow channel with high aff0inity. In-situ single crystal structures with the dose of $C_2H_2$, $CO_2$ or $C_2H_4$ further reveal the realistic host-guest interactions. Notably, rare $C_2H_2$ clusters are formed in the narrow channel while two distinct $CO_2$ adsorption locations are observed in the narrow channel and the large cavity with a ratio of 1:2, which accurately account for the distinct adsorption heat curves.

Ethylene ($C_2H_4$) is the foremost olefin as well as the highest volume product in the petrochemical industry, with an annual production capacity exceeding 214 million tons in 2021[1]. The manufacture of $C_2H_4$ and $C_3H_6$ accounts for 0.3% of global energy[2]. Current $C_2H_4$ production mainly relies on stream cracking of hydrocarbons[3–6]. Alternatively, oxidative coupling of methane ($CH_4$) has emerged as a promising technique to produce $C_2H_4$, among which acetylene ($C_2H_2$) and carbon dioxide ($CO_2$) are generated as byproducts and need to be deeply removed to produce polymer grade (>99.996%) $C_2H_4$[7]. Presently, multi-step purification process is adopted for purification of $C_2H_4$ from $C_2H_4$/$C_2H_2$/$CO_2$ mixtures in industry[8]. $C_2H_2$ is removed by catalytic hydrogenation using expensive noble-metal catalysts or solvent extraction, which is either energy intensive or associated with

pollution[9,10]. $CO_2$ is removed by chemical adsorption using caustic soda, which causes huge waste of costly solvents[11].

Physical adsorption offers potential to significantly reduce the energy footprint of separation processes[12–21]. Nonetheless, $C_2H_4$ purification from ternary $C_2H_4$/$C_2H_2$/$CO_2$ mixtures remains an unmet challenge due to the similarity in molecular size and polarity (Supplementary Table 2), although separation of $C_2H_2$/$C_2H_4$[22–26] or $C_2H_2$/ $CO_2$[27–32] binary mixtures has been realized by a plethora of porous materials. Besides, single-step purification of $C_2H_4$ from ternary $C_2H_2$/ $C_2H_4$/$C_2H_6$[33,34] or quaternary $C_2H_2$/$C_2H_4$/$C_2H_6$/$CO_2$[35] mixtures has also been realized by several porous materials. To date, less than ten materials have been demonstrated to separate $C_2H_4$ from $C_2H_4$/$C_2H_2$/ $CO_2$, including activated carbons, zeolites, covalent organic frameworks

[1]Key Laboratory of the Ministry of Education for Advanced Catalysis Materials, College of Chemistry and Life Sciences, Zhejiang Normal University, Jinhua 321004, China. [2]IBM Thomas J. Watson Research, Yorktown Heights, New York, NY 10598, USA. [3]Van't Hoff Institute for Molecular Sciences, University of Amsterdam, Science Park 904, 1098 XH Amsterdam, the Netherlands. [4]These authors contributed equally: Yunjia Jiang, Yongqi Hu. ✉e-mail: ybzhang@znu.edu.cn

and metal organic framework (MOFs)[36–39]. TIFSIX-17-Ni[36], NTU-65[37], and NTU-67[38] are so far the three optimal materials. TIFSIX-17-Ni[36] exhibits high $C_2H_2/C_2H_4$ and $CO_2/C_2H_4$ selectivity due to the negligible uptake of $C_2H_4$ under ambient condition. However, the capacity of $C_2H_2$ (3.30 mmol/g) and $CO_2$ (2.20 mmol/g) is relatively low due to the over-contracted channel. NTU-65[37] can selectively capture $C_2H_2$ and $CO_2$ by tuning the gate opening. However, the applied temperature must be at 263 K because lower temperatures lead to the adsorption of all the gases while higher temperatures cause the exclusion of $CO_2$. NTU-67[38] displays similar $C_2H_2$ (3.29 mmol/g) and $CO_2$ (2.04 mmol/g) capacity, but the $C_2H_2/C_2H_4$ and $CO_2/C_2H_4$ selectivity is greatly reduced as the $C_2H_4$ capacity (1.41 mmol/g) is relatively high. Additionally, the separation performance is deteriorated under humid conditions. Therefore, there is still a lack of ideal and stable materials to realize the simultaneous removal of $C_2H_2$ and $CO_2$ in $C_2H_2/CO_2/C_2H_4$ mixtures.

In this work, we reported a $GeF_6^{2-}$ anion embedded MOF ZNU-6 (ZNU = Zhejiang Normal University) with large cages (-8.5 Å diameter) connected by narrow interlaced channels (-4 Å diameter) for benchmark one-step $C_2H_4$ recovery from $C_2H_2$ and $CO_2$. ZNU-6 is constructed by $CuGeF_6$ and tri(pyridin-4-yl)amine (TPA) and exhibits excellent chemical stability. Static gas adsorption isotherms showed that ZNU-6 takes up 1.53/8.06 mmol/g of $C_2H_2$ and 1.46/4.76 mmol/g of $CO_2$ at 0.01 and 1.0 bar (298 K), respectively. The calculated IAST selectivities for $C_2H_2/C_2H_4$ (1/99) and $CO_2/C_2H_4$ (1/99) are 43.8–14.3 and 52.6–7.8 (0.0001–1.0 bar), respectively. The calculated $Q_{st}$ values at near-zero loading for $C_2H_2$ and $CO_2$ are 37.2 and 37.1 kJ/mol, indicative of its facility for material regeneration but much higher than that of $C_2H_4$ (29.0 kJ/mol). Modeling study indicates that there are two potential binding sites for $C_2H_2$, $C_2H_4$, and $CO_2$. One is in the interlaced channel and the other locates in the large cage. Moreover, all gas molecules prefer to be adsorbed in the interlaced channel with higher affinity. The realistic binding sites and host–guest interactions under normal conditions (298 K and 1.0 bar) were further demonstrated by in-situ single crystal structures with the saturated dose of gases. Notably, rare $C_2H_2$ clusters formed by π···π packing and C-H···C≡C interactions are observed in the interlaced channel with a small proportion of $C_2H_2$ molecules adsorbed in the large cage additionally. In sharp contrast, only 1/3 of $CO_2$ molecules are located in the narrow channel while 2/3 of $CO_2$ molecules are accommodated in the large cavity. This distinct gas distribution is highly consistent with the difference of adsorption heat curves. The practical $C_2H_4$ purification performance is further demonstrated by dynamic breakthroughs and record high $C_2H_4$ productivity is achieved from ternary $C_2H_2/CO_2/C_2H_4$ mixtures in a single adsorption process under various conditions. The separation performance is retained over multiple cycles and under humid conditions.

## Results

Violet single crystals of ZNU-6 (Supplementary Fig. 1) were produced by layering a MeOH solution of TPA onto an aqueous solution of $CuGeF_6$ (Fig. 1a). X-ray crystal analysis revealed that ZNU-6 [$Cu_6(GeF_6)_6(TPA)_8$] crystallizes in a three-dimensional (3D) framework in the cubic Pm-3n space group. Every unit cell consists of six $Cu^{2+}$ ions, six $GeF_6^{2-}$ anions, and eight tridentate TPA ligands (Supplementary Table 1). The combination of $Cu^{2+}$ and TPA produces a cationic pto network first (Fig. 1b), which determines the main pore size. The network is further embedded by $GeF_6^{2-}$ pillar to give a ith-d topology framework with optimal pore chemistry (Fig. 1c). The frameworks are composed of large icosahedral cage-like pores (-8.5 Å) and interlaced narrow channels (-4 Å) (Fig. 1d–f). Each large cage is surrounded by 12 channels and every interlaced channel connects 4 cages. The adjacent two cages and two channels share the same $GeF_6^{2-}$ anions at the edge. Both large pores and interlaced channels are abundant of Lewis basic F functional sites on the surface for gas binding. Such interconnected large cages and narrow channels are distinct from previous straight

1D channels of anion pillared MOFs (e.g., SIFSIX-1-Cu, SIFSIX-3-Ni). Besides, the narrow channel size may provide kinetic selectivity for $C_2H_2$ (3.3 Å) and $CO_2$ (3.3 Å) given their small molecular size compared to $C_2H_4$ (4.2 Å). Thus, ZNU-6 with abundant functional $GeF_6^{2-}$ binding sites, high porosity for $C_2H_2$ and $CO_2$ accommodation and narrow channel for kinetic preference features the promising characteristics for efficient purification of $C_2H_4$ from ternary $C_2H_2/CO_2/C_2H_4$ mixture.

The intrinsic porosity of ZNU-6 was investigated by $N_2$ adsorption at 77 K. As shown in Fig. 2a, ZNU-6 exhibited a type I adsorption isotherm. The Brunauer–Emmett–Teller surface area and pore volume were calculated to be 1330.3 $m^2$/g and 0.554 $cm^3$/g (Supplementary Fig. 10). The calculated pore size ranges from 8.22 to 10.76 Å with the summit in 9.0 Å, highly close to the pore aperture of -8.5 Å evaluated from the single crystal structure (Fig. 2a). Then, single-component adsorption isotherms of $C_2H_2$, $CO_2$, and $C_2H_4$ were collected at 298 K (Fig. 2b). At 1.0 bar, the $C_2H_2$ and $CO_2$ uptakes are 8.06 and 4.76 mmol/g, higher than those of most APMOFs (Fig. 2c). The capacities are equal to 4.68 and 2.77 gas molecules per $GeF_6^{2-}$ anion. Such high $C_2H_2$/anion and $CO_2$/anion uptakes have never been realized in anion pillared MOFs (Supplementary Table S7)[36–38,40–44]. Particularly, $C_2H_2$/anion and $CO_2$/anion uptakes in benchmark TIFSIX-17-Ni[36], SIFSIX-17-Ni[36] and NTU-67[38] are only 1.36/0.91, 1.29/0.9 and 2.06/1.28, respectively (Supplementary Fig. 25). So far, isomorphic SIFSIX-Cu-TPA[40] displays the ever highest $C_2H_2$/anion (4.44) uptake while SIFSIX-1-Cu[41] displays the ever highest $CO_2$/anion (2.72) uptake. It is worth mentioning that these records have been marginally surpassed by ZNU-6's (Supplementary Fig. 25). Notably, the uptakes of $C_2H_2$ and $CO_2$ on ZNU-6 at 0.01 bar are as high as 1.53 and 1.46 mmol/g, superior to those of all the porous materials in the context of ternary $C_2H_2/CO_2/C_2H_4$ separation, such as TIFSIX-17-Ni (1.38/0.32 mmol/g)[36], SIFSIX-17-Ni (0.91/0.20 mmol/g)[36], NTU-67 (0.47/0.65 mmol/g)[38], and TpPa-$NO_2$ (0.17/0.03 mmol/g)[39]. At 0.1 bar, the capacities of $C_2H_2$ and $CO_2$ reach up to 4.64 and 2.21 mmol/g (Fig. 2b), even higher than the uptakes of many porous materials at 1 bar and 298 K, for example, TIFSIX-17-Ni (3.30/2.20 mmol/g)[36]. In the meantime, the $C_2H_4$ uptakes on ZNU-6 at 0.01 and 0.1 bar are only 0.15 and 1.07 mmol/g, much lower than those of $C_2H_2$ and $CO_2$ under the same conditions. The $C_2H_2$, $CO_2$, and $C_2H_4$ adsorption isotherms were further collected at 278 and 308 K (Fig. 2d). The adsorption capacities of $C_2H_2$ and $CO_2$ at 1 bar increase to 8.74 and 6.26 mmol/g at 278 K. As selectivity is also an important parameter to assess the separation performance, we further calculated the $C_2H_2/C_2H_4$ and $CO_2/C_2H_4$ selectivities on ZNU-6 using ideal adsorbed solution theory (IAST) after fitting isotherms into dual site Langmuir or single site Langmuir equation with excellent accuracy. The IAST selectivity for 1/99 $C_2H_2/C_2H_4$ is 43.8–14.3 (Fig. 2e), higher than those of NTU-67 (8.1)[38] and TpPa-$NO_2$ (5.9)[39]. The IAST selectivities for 1/99 $CO_2/C_2H_4$ mixture is also as high as 52.6-7.8 (Fig. 2e). Besides, both $C_2H_2/C_2H_4$ and $CO_2/C_2H_4$ selectivity on ZNU-6 is improved with the pressure decrease or the increase of $C_2H_4$ ratios (from 90% to 99%) in the binary mixtures (Supplementary Figs. 13, 14), indicating ZNU-6 is favored for trace $C_2H_2$ and $CO_2$ capture from bulky $C_2H_4$ mixtures. Apart from the IAST selectivity, the Henry coefficients were also calculated to evaluate the Henry's selectivity of ZNU-6 (Supplementary Figs. 15–17), the Henry's selectivity for $C_2H_2/C_2H_4$ and $CO_2/C_2H_4$ is 8.2 and 7.8, respectively, superior to those of NTU-67 (2.4/4.2)[38] and TpPa-$NO_2$ (4.0/1.8)[39] (Supplementary Tables 4, 5). We further calculated the isosteric enthalpy of adsorption ($Q_{st}$) for ZNU-6 by using the Clausius-Clapeyron equation. $Q_{st}$ values at near-zero loading for $C_2H_2$, $CO_2$, and $C_2H_4$ are 37.2, 37.1, and 29.0 kJ/mol (Fig. 2f), respectively, indicative of the preferred affinity of $C_2H_2$ and $CO_2$ over $C_2H_4$. Notably, the $Q_{st}$ values for $C_2H_2$ and $CO_2$ on ZNU-6 are only modestly high and slightly lower than those of many top-performing materials in the context of $C_2H_4$ purification, such as SIFSIX-17-Ni (44.2/40.2 kJ/mol)[36], TIFSIX-17-Ni (48.3/37.8 kJ/mol)[36], and NTU-67 (44.1/41.5 kJ/mol)[38]. Such moderate $Q_{st}$ endows facile regeneration of ZNU-6 under mild conditions.

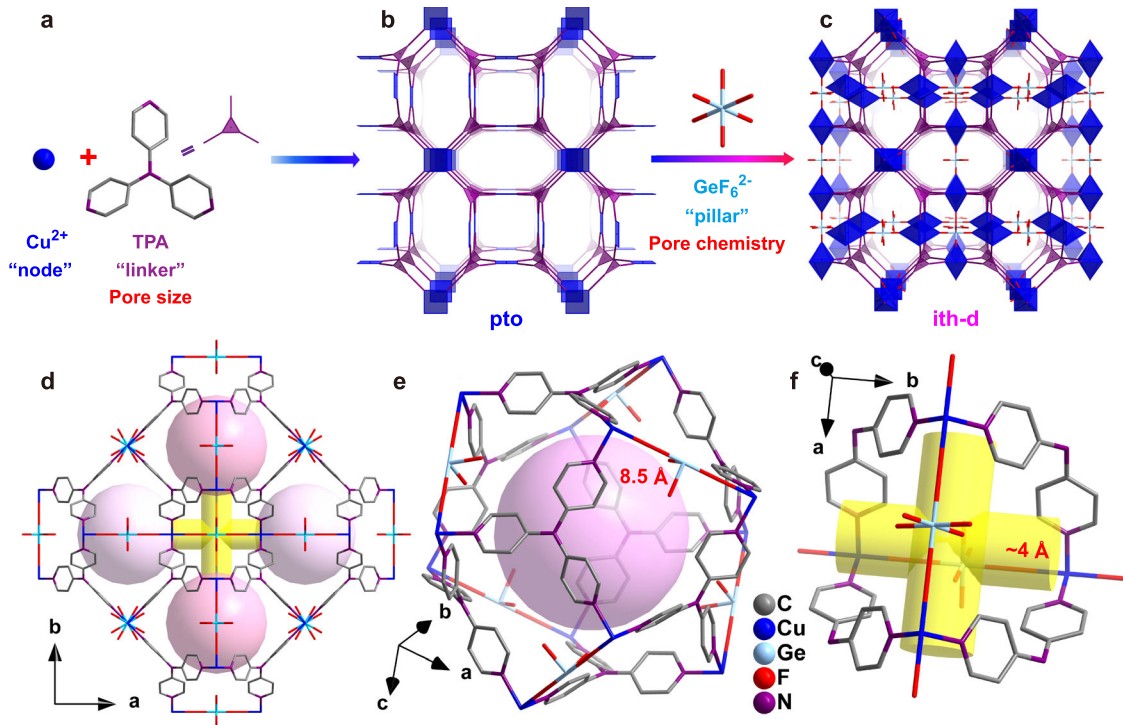

**Fig. 1 | Porous structure of ZNU-6. a–c** Exquisite control of pore size/shape and pore chemistry in ZNU-6 from pillared (3,4)-connected pto network to $GeF_6^{2-}$ embedded ith-d topology framework; **d** Overview of ZNU-6 structure with cage-like pores and interlaced channels. **e** Structure and size of the cage-like pore. **f** Structure and size of the interlaced channel connecting four cages.

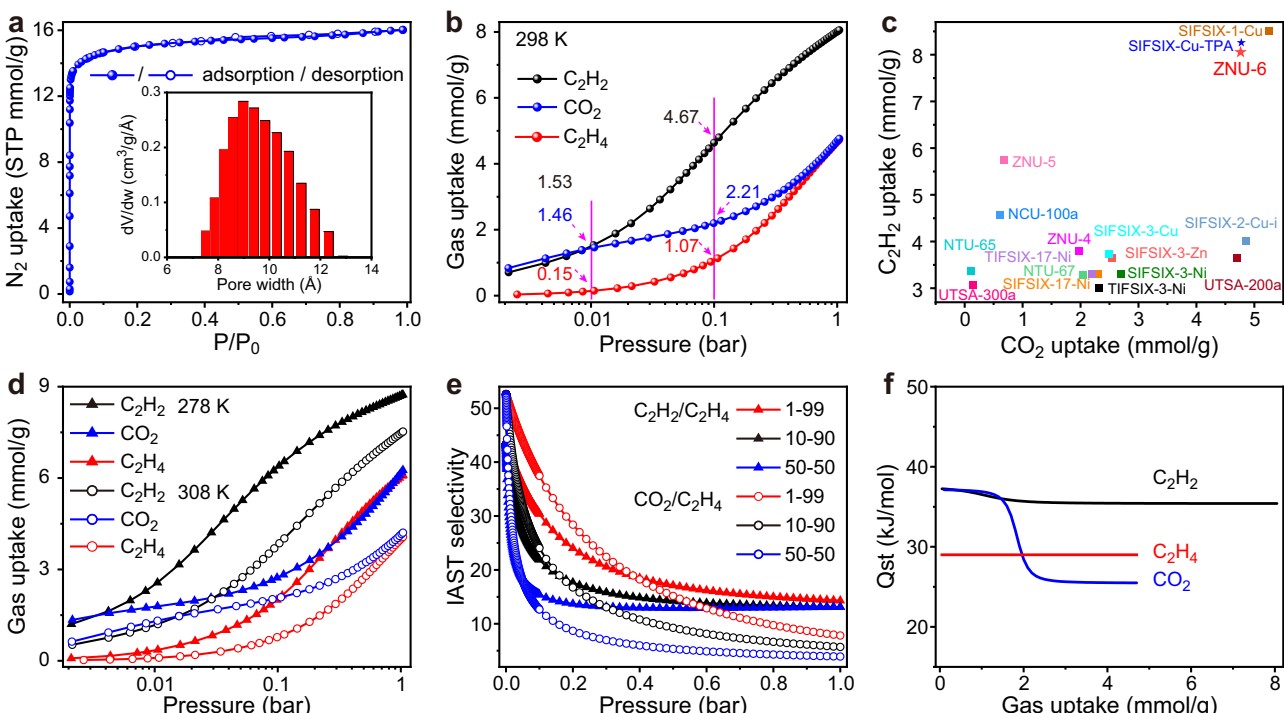

**Fig. 2 | The sorption performance. a** $N_2$ adsorption and desorption isotherms for ZNU-6 and the calculated pore size distribution. **b** $C_2H_2$, $CO_2$, and $C_2H_4$ adsorption isotherms of ZNU-6 at 298 K. **c** Comparison of the saturated $C_2H_2$ and $CO_2$ uptake (1 bar, 298 K) among anion pillared MOFs. **d** $C_2H_2$, $CO_2$, and $C_2H_4$ adsorption isotherms of ZNU-6 at 278/308 K. **e** $C_2H_2/C_2H_4$ and $CO_2/C_2H_4$ IAST selectivity of ZNU-6 at 298 K. **f** $Q_{st}$ of $C_2H_2$, $CO_2$, and $C_2H_4$ in ZNU-6. Source data are provided as a Source Data file.

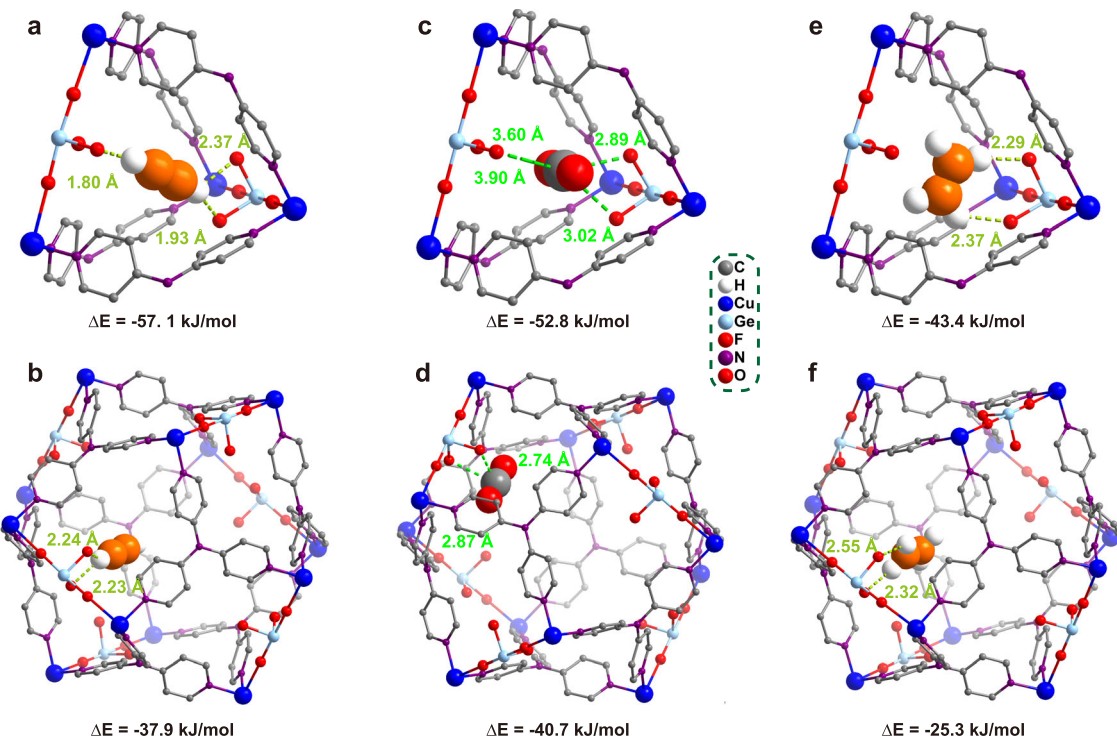

**Fig. 3 | The DFT optimized gas adsorption configuration.** Binding sites I, II of $C_2H_2$ (**a, b**), $CO_2$ (**c, d**), and $C_2H_4$ (**e, f**).

To gain more insights into the gas adsorption behavior, density functional theory (DFT)-based calculations (see Method section) were applied to identify the adsorption configuration and binding energies of $C_2H_2$, $CO_2$, and $C_2H_4$. For all gases, two different binding sites were observed. Site I is in the interlaced channel and Site II is in the large cavity (Fig. 3). For $C_2H_2$ in Site I, the two hydrogen atoms interact strongly with three F atoms with the distances of 1.80, 1.93, and 2.37 Å. The calculated binding energy is 57.1 kJ/mol (Fig. 3a). As for $C_2H_2$ adsorbed in Site II, only one hydrogen atom can interact with the adjacent F atoms with the distance of 2.23 and 2.24 Å, and the corresponding binding energy decreases to 37.9 kJ/mol (Fig. 3b), indicating that $C_2H_2$ is preferentially adsorbed in the narrow channel. The same results are also observed for $CO_2$ and $C_2H_4$, the binding energies in the channel are much higher than those in the large cage. In Site I, $CO_2$ is trapped by two strong and two weak electrostatic F⋯C=O interactions in the distance of 2.89, 3.02, 3.60, and 3.90 Å, the binding energy is 52.8 kJ/mol (Fig. 3c); $C_2H_4$ is adsorbed via two F⋯H interactions (2.29 and 2.37 Å) with the binding energy of 43.3 kJ/mol (Fig. 3e). In Site II, the binding energy of $CO_2$ drops to 40.7 kJ/mol with the number of electrostatic F⋯C=O interactions (2.74 and 2.87 Å) decreasing to two (Fig. 3d); the binding energy of $C_2H_4$ reduces to 25.3 kJ/mol with the length of F⋯H extending to 2.55 and 2.32 Å (Fig. 3f). In addition, it is notable that either in Site I or II, the binding energy of $C_2H_2$ or $CO_2$ is superior to that of $C_2H_4$, confirming that the adsorption of $C_2H_2$ or $CO_2$ in ZNU-6 is more preferable than that of $C_2H_4$.

Although DFT calculations have identified two different binding sites for each gas, it is still difficult to understand the distinct adsorption heat curves. Therefore, we further studied the in-situ structures of ZNU-6 with gas loading (Fig. 4). We found that averagely 25.78 $C_2H_2$, 18 $CO_2$, or 13.07 $C_2H_4$ molecules can be adsorbed per unit cell of ZNU-6 (Supplementary Table 1), corresponding to 4.3 $C_2H_2$, 3.0 $CO_2$, and 2.2 $C_2H_4$ molecules for each $GeF_6^{2-}$ anion, which are close to the saturated values from gas adsorption isotherms (4.63 $C_2H_2$, 2.77 $CO_2$, and 2.75 $C_2H_4$). Both of $C_2H_2$ and $CO_2$ have two binding sites, i.e., Site I in the interlaced channel and Site II in the large cage. Notably, the amount of $C_2H_2$ molecules distributed to the two locations is 3.8 and

0.5 per $GeF_6^{2-}$ anion while that for $CO_2$ is 1 and 2 per $GeF_6^{2-}$ anion (Fig. 4a, b). Such different gas distribution can precisely account for the $C_2H_2$ $Q_{st}$ curve with a modest decrease and the $CO_2$ $Q_{st}$ curve with a sharp decrease along the gas loading. Specifically, $C_2H_2$ molecules adsorbed in Site I bind to F atoms on the surface of the channels via multiple cooperative hydrogen bonds (C-H⋯F = 1.97–2.55 Å), and the others in Site II interact F atoms via single H⋯F hydrogen bond with the distance of 2.51 Å (Fig. 4a and Table 1). Besides, the $C_2H_2$ molecules in Site I aggregate to form a stacked gas cluster by π⋯π packing and C-H⋯C≡C interactions, which has rarely been observed previously. Regarding $CO_2$, it is trapped by F⋯C=O electrostatic interaction in Site I and II (Fig. 4b). The only difference is that the C⋯F distance is 2.64 Å in Site I and 2.80 Å in Site II (Table 1). From the single crystal structure, two different $CO_2$ molecules that are very close and opposite to each other in the narrow channel (site I) are observed. However, these two $CO_2$ molecules cannot exist in the same narrow channel at the same time and thus both $CO_2$ molecules display the occupancy of 50%. In Site II, the C atom of $CO_2$ is ordered while the O atoms are disordered to two perpendicular positions with the occupancy of 50% for each configuration. Besides, the linear $CO_2$ molecules are slightly bent due to the strong attraction from $GeF_6^{2-}$ anion. The bent angle of 157.5° (in Site I) and 170.8° (in Site II) are consistent with the interaction strength. In term of $C_2H_4$, only one site in the narrow channel is found. The C atoms of $C_2H_4$ molecule are ordered while the H atoms are disordered. The distances of C-H⋯F interactions between $C_2H_4$ and framework are 2.31–2.64 Å (Fig. 4c and Table 1). Considering the slight lower $C_2H_4$/ $GeF_6^{2-}$ ratio observed in the single crystal structure, there should be some $C_2H_4$ molecules adsorbed in the large cage. However, due to the probable disorder of $C_2H_4$ molecules over the whole cage, the $C_2H_4$ molecules in Site II were not solved. Nonetheless, this uniform adsorption configuration is consistent with the flat $Q_{st}$ curve for $C_2H_4$.

Apart from the $C_2H_2$, $CO_2$, or $C_2H_4$ molecules, some water molecules were also identified in the framework (Supplementary Fig. 4). As there is still a lot of space in the large cavity after saturated adsorption of $C_2H_2$, $CO_2$, or $C_2H_4$ gases at 100 kPa, the water adsorption behavior probably occurred during the single crystal measurement, which is

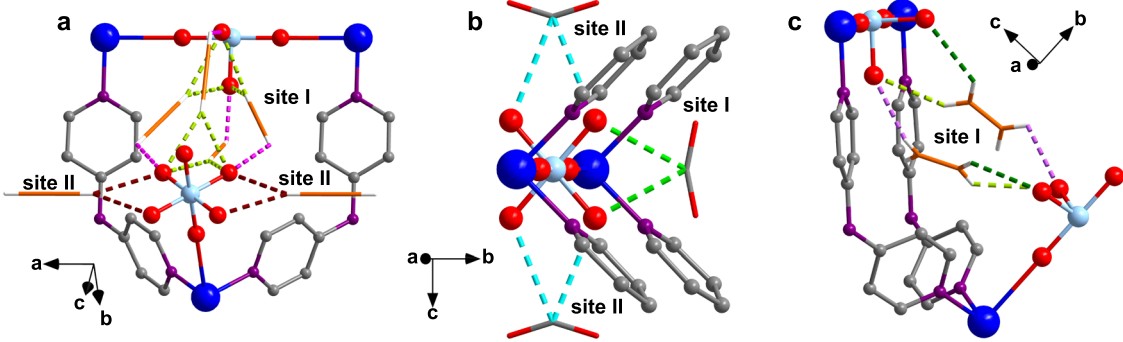

**Fig. 4 | Single crystal structure of gas-loaded ZNU-6. a** $C_2H_2$ @ ZNU-6 [$Cu_6(GeF_6)_6(TPA)_8$ $(C_2H_2)_{25.78}$]; **b** $CO_2$ @ ZNU-6 [$Cu_6(GeF_6)_6(TPA)_8$ $(CO_2)_{18}$]; **c** $C_2H_4$ @ ZNU-6 [$Cu_6(GeF_6)_6(TPA)_8$ $(C_2H_4)_{13.07}$].

exposed to air. Interestingly, these water molecules are distant from $GeF_6^{2-}$, indicating that these $H_2O$ molecules do not occupy the binding sites for the targeted gases. Instead, some unique interactions are observed between the gas molecules and water molecules, e.g., O-H···C=O hydrogen bonds between $CO_2$ and $H_2O$. Notably, our resolved single crystal structures show completely different $C_2H_2$ and $CO_2$ adsorption configurations from those of isomorphic SIFSIX-Cu-TPA for $C_2H_2/CO_2$ separation in Wu's work[39].

Motivated by the high adsorption capacity and selectivity in single-component adsorption as well as the in-situ single crystal structure analysis, breakthrough experiments were conducted for $C_2H_2/C_2H_4$, $CO_2/C_2H_4$, and $C_2H_2/CO_2/C_2H_4$ mixtures. The results showed that highly efficient separations can be accomplished by ZNU-6 for all the gas mixtures under various conditions. For 1/99 $C_2H_2/C_2H_4$ mixtures, $C_2H_4$ is eluted at 12 mins while $C_2H_2$ is detected until 192 min. For 10/90 $CO_2/C_2H_4$ mixtures, $C_2H_4$ and $CO_2$ are detected at 12 and 43.5 min, respectively (Fig. 5a). For 1/1/98 $C_2H_2/CO_2/C_2H_4$ mixtures, $C_2H_2$ and $CO_2$ broke out simultaneously and 64.42 mol/kg of polymer grade $C_2H_4$ is produced by single adsorption process (Fig. 5b). The productivity is improved to 80.89 mol/kg when decreasing the temperature to 283 K (Supplementary Fig. 46). The $CO_2$ breakthrough time becomes shortened with the increase of $CO_2$ ratio, which is 72 and 52 min for 1/5/94 (Figs. 5c) and 1/9/90 (Fig. 5d) $C_2H_2/CO_2/C_2H_4$ mixtures. The polymer grade $C_2H_4$ productivity is 21.37 and 13.81 mol/kg, respectively. As most reported $C_2H_4$ productivity from $C_2H_2/CO_2/C_2H_4$ mixtures are compared under 1/9/90, a comparison plot of the $C_2H_4$ productivity and dynamic $C_2H_2$ capacity from 1/9/90 $C_2H_2/CO_2/C_2H_4$ mixtures is presented in Fig. 5e. ZNU-6 displays the record high $C_2H_4$ productivity and second highest $C_2H_2$ dynamic capacity. The $C_2H_4$ productivity of ZNU-6 is >2.5 folds of the previous benchmark of NTU-67 (5.42 mol/kg)[38]. $C_2H_4$ productivity with the unit of mol/kg/h is also calculated for comparison (Supplementary Table S10). ZNU-6 with the productivity of 15.93 mol/kg/h is the highest reported value.

In view of the importance of the recyclability and stability of porous materials for practical applications, the water and thermal stability of ZNU-6 was investigated. There was no noticeable loss in the $CO_2$ adsorption capacity after six cycles of adsorption/desorption experiments (Supplementary Fig. 26). Long time soaking of ZNU-6 in water or polar organic solvents such as DMSO, DMF and MeCN did not change the porous structure of ZNU-6, as demonstrated by the PXRD

patterns as well as the gas adsorption isotherms (Supplementary Fig. 7). Thermogravimetric analysis (TGA) and temperature varied PXRD indicated ZNU-6 is stable below 200 °C (Supplementary Figs. 8, 9). Breakthroughs under humid conditions or over four cycles preserved nearly the identical separation performance (Fig. 5f). Although many water molecules can be adsorbed in ZNU-6, as described in in-situ crystals and water adsorption isotherms (Supplementary Fig. 27), the presence of humid has negligible influence on the separation performance (Fig. 5f). This is probably due to the co-adsorption of water and target gases as well as the fast $C_2H_2/CO_2/C_2H_4$ diffusion kinetics (Supplementary Fig. 29–31).

## Discussion
In conclusion, we reported a $GeF_6^{2-}$ anion embedded metal organic framework ZNU-6 with optimal pore structure and pore chemistry for benchmark one-step $C_2H_4$ purification by simultaneous removal of $C_2H_2$ and $CO_2$. ZNU-6 exhibits remarkably high $C_2H_2$ and $CO_2$ capacity under both low and high pressures. The $C_2H_2$/anion and $CO_2$/anion uptakes are the highest among all the anion pillared MOFs. 64.42, 21.37, 13.81 mol/kg polymer grade $C_2H_4$ can be produced from $C_2H_2/CO_2/C_2H_4$ (1/1/98, 1/5/94, 1/9/90) mixtures, all superior to the previous benchmarks. The separation performance is sustained over multiple cycles or under humid conditions. The potential gas binding sites are investigated by DFT calculation, which indicate that $C_2H_2$ and $CO_2$ are preferentially adsorbed in the interlaced narrow channel with high affinity. In-situ single crystal structures with the dose of $C_2H_2$, $CO_2$ or $C_2H_4$ further reveal the realistic host–guest interactions, accounting for the distinct shapes of the adsorption heat curves. In general, our work highlights the significance of regulating pore structure and pore chemistry in porous materials to construct multiple cooperative functionalities for gas separation.

## Methods
### Synthesis of ZNU-6
To a 5 mL long thin tube was added a 1 mL of aqueous solution with $Cu(NO_3)_2·3H_2O$ (~1.3 mg) and $(NH_4)_2GeF_6$ (~1.0 mg). 2 mL of $MeOH/H_2O$ mixture (v:v = 1:1) was slowly layered above the solution, followed by a 1 mL of MeOH solution of TPA (~1.0 mg). The tube was sealed and left undisturbed at 298 K. After ~1 week, purple single crystals were obtained.

### Preparation of gas loaded ZNU-6
The crystalline sample of ZNU-6 was filled into a glass tube and heated at 120 °C under vacuum for 24 h. After the sample cooling down, $CO_2$, $C_2H_2$, or $C_2H_4$ was introduced into the sample respectively with Builder SSA 7000 (Beijing) instrument until the pressure reach to 1 bar at 298 K and the state is maintained for another hour. Then, the crystals were picked out, covered with the degassed oil, and single crystal X-ray diffraction measurements were then carried out at 298 K as soon as possible.

**Table 1 | The distances of the host–guest interactions**

| Crystals | Site I | Site II |
|---|---|---|
| 25.78 $C_2H_2$ @ ZNU-6 | 1.97/2.55 Å (C-H···F) | 2.51 Å (C-H···F) |
| 18 $CO_2$ @ ZNU-6 | 2.64 Å (C···F) | 2.80 Å (C···F) |
| 13.07 $C_2H_4$ @ ZNU-6 | 2.31/2.54/2.64 Å (C-H···F) | - |

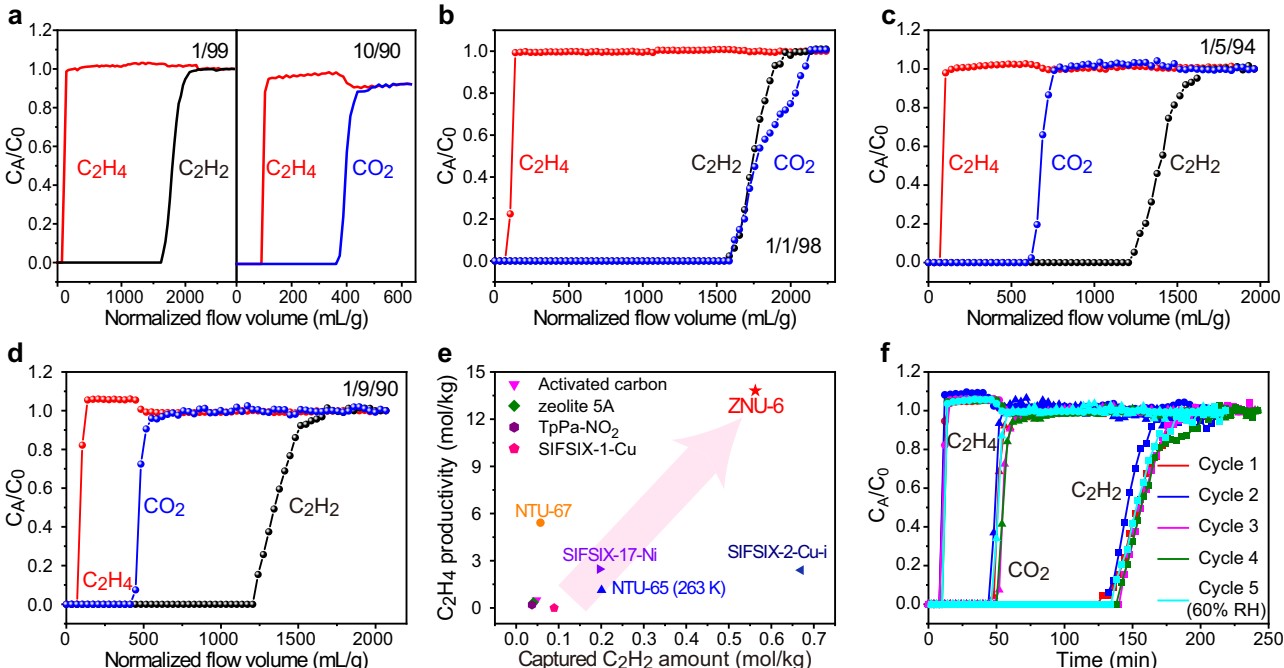

**Fig. 5 | C₂H₄ purification.** Experimental breakthrough curves of ZNU-6 for binary mixture **a** C₂H₂/C₂H₄ (1/99) and CO₂/C₂H₄ (10/90) at 298 K. Experimental breakthrough curves of ZNU-6 for ternary mixture **b** C₂H₂/CO₂/C₂H₄ (1/1/98), **c** C₂H₂/CO₂/C₂H₄ (1/5/94), and **d** C₂H₂/CO₂/C₂H₄ (1/9/90). **e** Comparison of the captured C₂H₂ amount and C₂H₄ productivity from C₂H₂/CO₂/C₂H₄ (1/9/90) ternary mixture. **f** Five cycles of experimental breakthrough curves of ZNU-6 for C₂H₂/CO₂/C₂H₄ (1/9/90) at 298 K (1–4: dry condition, 5: humid condition). Source data are provided as a Source Data file.

### Single-crystal X-ray diffraction
Single-crystal X-ray diffraction studies were conducted on the BrukerAXS D8 VENTURE diffractometer equipped with a PHOTON-100/CMOS detector (GaKα, $\lambda = 1.34139$ Å). Indexing was performed using APEX2. Data integration and reduction were completed using SaintPlus 6.01. Absorption correction was performed by the multiscan method implemented in SADABS. The space group was determined using XPREP implemented in APEX2.1 The structure was solved with SHELXS-97 (direct methods) and refined on F2 (nonlinear least-squares method) with SHELXL-97 contained in APEX2, WinGX v1.70.01, and OLEX2 v1.1.5 program packages. All non-hydrogen atoms were refined anisotropically. The contribution of disordered solvent molecules was treated as diffuse using the Squeeze routine implemented in Platon.

### Powder X-ray diffraction
Powder X-ray diffraction (PXRD) data were collected on the SHIMADZU XRD-6000 diffractometer (Cu Kα$\lambda = 1.540598$ Å) with an operating power of 40 kV, 30 mA and a scan speed of 4.0°/min. The range of 2θ was from 5° to 50°.

### Thermal gravimetric analysis
Thermal gravimetric analysis was performed on the TGA STA449F5 instrument. Experiments were carried out using a platinum pan under nitrogen atmosphere which conducted by a flow rate of 60 mL/min nitrogen gas. The data were collected at the temperature range of 50 °C to 600 °C with a ramp of 10 °C /min.

### The static gas/vapor adsorption equilibrium measurements
The static gas adsorption equilibrium measurements were performed on the Builder SSA 7000 instrument. The water vapor adsorption equilibrium measurements were performed on the BeiShiDe DVS instrument. Before measurements, the sample of ZNU-6 (-100 mg) was evacuated at 25 °C for 2 h firstly, and then at 120 °C for 12 h until the pressure dropped below 7 μmHg. The sorption isotherms were collected at 77 K, 278, 298, and 308 K on activated samples. The experimental temperatures were controlled by liquid nitrogen bath (77 K) and water bath (278, 298, and 308 K), respectively.

### Breakthrough experiments
The breakthrough experiments were carried out on a dynamic gas breakthrough equipment. The experiments were conducted using a stainless steel column (4.6 mm inner diameter × 50 mm length). The weight of ZNU-6 powder packed in the columns were 0.5806 g. The column was activated at 75 °C for 2 h under vacuum, and then raised to 120 °C for overnight. The mixed gas of C₂H₂/C₂H₄ (1/99, v/v), CO₂/C₂H₄ (10/90, v/v), or C₂H₂/CO₂/C₂H₄ (1/9/90, 1/5/94, 1/1/98, 5/5/90, v/v/v) was then introduced. C₂H₂/CO₂/C₂H₄ mixtures are produced by mixing three pure gases or mixing binary mixture with pure gas. Every flowrate was calibrated by self-made soap film flowmeter. Outlet gas from the column was monitored using gas chromatography (GC-9860-5CNJ) with the thermal conductivity detector TCD. After the breakthrough experiment, the sample was regenerated with an Ar flow of 5 mL min⁻¹ under 120 °C for 8 h or under vacuum at 120 °C for 8 h.

### Fitting of experimental data on pure component isotherms
The unary isotherms for C₂H₂ and CO₂ measured at three different temperatures 278 K, 298 K, and 308 K in ZNU-6 were fitted with excellent accuracy using the dual-site Langmuir model, where we distinguish two distinct adsorption sites A and B:

$$q = \frac{q_{sat,A}b_Ap}{1+b_Ap} + \frac{q_{sat,B}b_Bp}{1+b_Bp} \tag{1}$$

In Eq (S1), the Langmuir parameters $b_A, b_B$ are both temperature dependent

$$b_A = b_{A0}\exp\left(\frac{E_A}{RT}\right); b_b = b_{B0}\exp\left(\frac{E_B}{RT}\right) \tag{2}$$

In Eq. (2), $E_A, E_B$ are the energy parameters associated with sites A, and B, respectively.

The corresponding unary isotherms for $C_2H_4$ measured at three different temperatures 278 K, 298 K, and 308 K in ZNU-6 were fitted with excellent accuracy using the single-site Langmuir model.

$$q = \frac{q_{sat,A}b_A p}{1+bp} \quad (3)$$

The unary isotherm fit parameters for $C_2H_2$, $CO_2$, and $C_2H_4$ are provided in Table S1.

## IAST calculations

The adsorption selectivity for separation of binary mixtures of species 1 and 2 is defined by

$$S_{ads} = \frac{q_1/q_2}{p_1/p_2} \quad (4)$$

where $q_1$, $q_2$ are the molar loading (units: mol kg$^{-1}$) in the adsorbed phase in equilibrium with a gas mixture with partial pressures $p_1$, $p_2$ in the bulk gas.

## Calculation of isosteric heat of adsorption ($Q_{st}$)

The isosteric heat of adsorption, $Q_{st}$, is defined as

$$Q_{st} = -RT^2 \left( \frac{\partial \ln p}{\partial T} \right)_q \quad (5)$$

where the derivative in the right member of Eq. (5) is determined at constant adsorbate loading, $q$. The calculations are based on the Clausius-Clapeyron equation.

## Density functional theory calculation

In this work, the DFT-based calculations were carried out using the CP2K package[45]. The Perdew-Burke-Ernzerhof (PBE) exchange functional[46], Gaussian plane wave (PAW) pseudopotentials[47] and DZVP basis sets[48] for carbon, oxygen, fluorine, nitrogen, germanium and copper atoms, were used to describe the exchange–correlation interactions and electron–ion interaction, respectively. At the same time, the PBE-D3 method[49] with Becke–Jonson damping for all atoms and Hubbard U corrections for the open-shell 3d transition metal (Cu) was used for geometry optimizations. The $U$ value of 5.0 eV was used in this study. In all calculations, the net charges of simulation systems were set to zero. The adsorption energy can be obtained from formula below:

$$E_{ads} = E_{adsorbate+substrate} - E_{substrate} - E_{adsorbate} \quad (6)$$

where $E_{adsorbate+substrate}$ and $E_{substrate}$ were the total energies of the substrate with and without adsorbate, and $E_{adsorbate}$ was the energy of the adsorbate.

## Data availability

The authors declare that the data supporting the findings of this study are available within the article and Supplementary Information. The X-ray crystallographic data related to ZNU-6 have been deposited at the Cambridge Crystallographic Data Centre (CCDC), under deposition numbers 2192744–2192747, respectively. These data can be obtained free of charge from the CCDC via www.ccdc.cam.ac.uk/data_request/cif. The data that support the findings of this study are available from the corresponding author. Besides, Source data are provided with this paper.

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

## Acknowledgements

This work was supported by the National Natural Science Foundation of China (Nos. 21908193 and 22205207) and Jinhua Industrial Key Project (2021A22648). We thank the help of Dr. Yunlei Peng from CUP.

## Author contributions

Y.J. and Y.H. contributed equally to this work. Y.Z. designed and guided the project. Y.J. and Y.H. designed and synthesized the materials, performed the majority of the structural characterization, collected gas sorption data and conducted breakthrough experiments. B. L. conducted the DFT calculations. L.W. and H.N. collected X-ray diffraction data and solved the structures. R.K. performed the IAST and $Q_{st}$ calculation. Y.J. and Y.Z. draft the paper. X.H. provided important advice and revised the paper. All authors contributed to the discussion of results and commented on the paper.

## Competing interests

The authors declare no competing interests.
