## [Peer Review File · Nature Communications]

Benchmark Single-Step Ethylene Purification from Ternary Mixtures by a Customized Fluorinated Anion Embedded MOFREVIEWER COMMENTS

Reviewer #1 (Remarks to the Author):

In this work authors reported GeF62- anion incorporated MOF named ZNU-6. The MOF has optimum pore structure and environment for one step ethylene purification and also high selective for carbon di oxide separation as well. This research area is very promising epically today when we need to find better alternative for energy storage and separation. The authors investigated the MOF in different conditions and proved its viability and effectiveness. The chemical approach of this work is strong, and the results are clearly visible from the experiments

However here are some of my questions and concerns

1. The topic of ethylene purification/ separation using MOF is not very new, even incorporation of anions in MOFs has been reported by the authors group recently (Wang et al, Nano Research 2022). It would be great if authors can describe the novelty of this research as suitable of Nature common
2. Its seems that the customized pore environment helps to help achieve higher selectivity however DFT studies, and in-situ X ray have been performed to corroborate the statement. Have the authors performed any in-situ IR to verify the conclusions? This evidence will help strengthen the case
3. Heat of adsorption values and isotherm dependency with temp indicates the physisorption nature of guest /host chemistry. However, the additional interaction with anions and even TPA can provide some chemisorption in CO2. Have the authors performed any Temp programed desorption studies in terms of gases of choice to see the binding behavior?
4. What is the relative humidity used in this study?
5. We can see ZNU-6 is strong water adsorbent however in humid conditions shows negligible deterioration of separation performance. However, the reasoning and data is not very clear, has the authors tried to quantify the effect of humidity in this case,
6. In similar topic, what happen in desorption cases in breakthrough, I mean when the fixed bed is undergoing regeneration, have the authors quantified the effect of humidity in gas desorption cases and in cyclic performance?

Reviewer #2 (Remarks to the Author):

The authors reported a GeF62- anion embedded MOF, ZNU-6, with optimized pore structure and environment for highly efficient C2H4 recovery from various C2H2/CO2/C2H4 ternary mixtures. The material exhibited good recyclability and resistance toward moisture during breakthrough experiments. Ethylene (C2H4) purification from multi-component mixtures by physical adsorption is an important industrial challenging, and this work will be very interesting for the readers. The experiments were well performed and the manuscript has been written well. I agree that this manuscript to be accepted for Nature Communications after some minor revision. The following are the suggestions for the authors to improve the manuscript.

1. Figure 1 Caption, (f) is inconsistent with other numbering.
2. Supplementary Figure 10-11, the authors should provide the pressure of these IAST selectivities.
3. For the in-situ gas-loaded MOF structure, the authors should add a figure to describe the C2H2 gas molecule cluster and the interactions between the molecules, either in main text or SI. A figure illustrating the interaction between adsorbed water and gas molecules should also be added.

Reviewer #3 (Remarks to the Author):

Thanks to its potential to introduce energy-efficient, single-step ethylene purification

approaches, this manuscript by Jiang et al. is of high topical relevance to gas purifications, and MOFs for separations. Considering high importance of the new findings, and their general relevance in controlling the pore electrostatics (F...C=O interactions) driven gas separation/purification properties, I support publication subject to necessary revisions as follows:

- Molecular formulae are missing including that of the new as-synthesised ZNU-6 (although I find this formula in Table S7, should be clearly noted in the manuscript too). All molecular formulae for the gas-loaded phases should be clearly written in the main article, if needed, using a table with analysis of sorbate-sorbent interactions (distances).

- The authors should not be presenting the adsorption data with respect to molecules per GeF6²⁻ anion, this makes the data skewed in favour of this adsorbent, ZNU-6. Also, the units for adsorption uptakes and pressure need to be consistent. Right now, with mixed use of cm³/g, mmol/g, mol/mol and mmol/g, these are all mixed up; same mix-up is observed in the units used for pressure, bar and kPa. Collectively, all these mistakes come together in the manuscript Figure 2. I strongly recommend replotting the isotherm-based uptakes in one consistent pair of units: mmol/g and bar. This not only makes this manuscript coherent but also helps the whole community working in this area with regard to comparing performance parameters across different adsorbents.

- Page 2, lines 33-34, the authors need to discuss "Presently, multi-step purification process is adopted for purification of C₂H₄ from C₂H₄/C₂H₂/CO₂ mixtures." This needs to take cognisance of the recent reports on single-step C₂H₄ purification from ternary (1:1:1 for C₂H₂/C₂H₄/C₂H₆) and quaternary (1:1:1:1 for C₂H₂/C₂H₄/C₂H₆/CO₂) gas mixtures. Some examples include: Cao, JW. et al., Nat Commun. 2021, 12, 6507. Xu, Z. et al., Nat Commun. 2020, 11, 3163.

- I don't quite agree with the authors use of the word "static" in the heading of Figure 2. Should just be written as "The sorption performance." In another instance, the authors use this phrase "static adsorption" (page 11, line 202), which I object to.

- It is stated that the C₂H₄ productivity is 309 mL/g, again this unit is far from the standard unit typically used in other literature reports in this area, i.e., mol kg⁻¹ h⁻¹.

- Importantly, the authors have calculated productivity by integrating the effluent flow rate of C₂H₄ (cm³/min). However, all the breakthrough curves (in Fig. 5) are showed in with the effluent concentration (C/C₀). It is obvious that integrating the concentration curve cannot give the amount, although a few references adopt this wrong method. Therefore, it is necessary to show the details for direct measurement and/or indirect calculation of the effluent flow rate and concentration in this study. See Shen, J. et al., Nat Commun. 2020, 11, 6259 as a reference.

- How common / rare is the it-h topology among the anion-pillared MOF library? It would greatly help the article / future readers, if the authors can address this by a thorough literature-based contextualisation of the structural topology of ZNU-6.

- Section 7 in the supplementary information has a header "Breakthrough simulations and experiments", the following figure captions for Supplementary Figures 28-43 are all including "Experimental" in their figure captions, nonetheless. I wonder where the simulation-derived breakthrough data is?

Overall, an interesting idea executed by the authors that should advance this area in the near future. I will be glad to look at a suitably revised article, when ready.

Reviewer #4 (Remarks to the Author):

This manuscript by Jiang et al, report a metal organic framework (ZNU-6) for the application of simultaneous removal of C₂H₂ and CO₂ from C₂H₄ stream. This area is of important practical applications and has been extensively investigated in the past few years. The overall quality of this manuscript is high with detailed investigation of the structure characterisation of the MOF, study of its adsorption properties using isotherm and breakthrough experiments, and thorough investigation of the host-guest interactions. Below are some comments and questions from me, and I would recommend the publication of this manuscript after the authors fully address them:

1. I think it is necessary that the authors reference some highly relevant publications in

the introduction, e.g. Hexafluorogermanate (GeFSIX) Anion-Functionalized Hybrid Ultramicroporous Materials for Efficiently Trapping Acetylene from Ethylene, *Ind. Eng. Chem. Res.* 2018, 57, 21, 7266–7274.

2. The cif. File for CO₂ loaded ZNU-6 shows an O-C-O bond angle of CO₂ being 157 ° instead of the theoretical 180°, could the authors please explain this discrepancy.

3. Figure 2f, the Q_{st} for CO₂, why it showed a sudden drop at 2mmol/g coverage?

4. From the isotherms for CO₂ and C₂H₄ (Figure 2b, 2d), the saturation uptake for CO₂ and C₂H₄ are very close; and the kinetic data (supplementary figure 25) of CO₂ and C₂H₄ are also similar with C₂H₄ being slightly faster in adsorption. For Q_{st}, CO₂ started higher than C₂H₄ then fall to be lower than C₂H₄. These three parameters (uptake, kinetic, Q_{st}), all seem indicating the interaction between the gas molecules and the framework is very similar. In this case, how do the authors rationalise the observed separation in breakthrough experiments? what do the authors think is the really reason that ZNU-6 can retain CO₂ from mixtures containing mainly C₂H₄ and small percentage of CO₂.

REVIEWER COMMENTS

Reviewer #1 (Remarks to the Author)

In this work authors reported GeF_6^{2-} anion incorporated MOF named ZNU-6. The MOF has optimum pore structure and environment for one step ethylene purification and also high selective for carbon dioxide separation as well. This research area is very promising especially today when we need to find better alternative for energy storage and separation. The authors investigated the MOF in different conditions and proved its viability and effectiveness. The chemical approach of this work is strong, and the results are clearly visible from the experiments

However here are some of my questions and concerns

1. The topic of ethylene purification/ separation using MOF is not very new, even incorporation of anions in MOFs has been reported by the authors group recently (Wang et al, Nano Research 2022). It would be great if authors can describe the novelty of this research as suitable of Nature common.
2. Its seems that the customized pore environment helps to achieve higher selectivity however DFT studies, and in-situ X ray have been performed to corroborate the statement. Have the authors performed any in-situ IR to verify the conclusions? This evidence will help strengthen the case
3. Heat of adsorption values and isotherm dependency with temp indicates the physisorption nature of guest /host chemistry. However, the additional interaction with anions and even TPA can provide some chemisorption in CO_2 . Have the authors performed any Temp programed desorption studies in terms of gases of choice to see the binding behavior?
4. What is the relative humidity used in this study?
5. We can see ZNU-6 is strong water adsorbent however in humid conditions shows negligible deterioration of separation performance. However, the reasoning and data is not very clear, has the authors tried to quantify the effect of humidity in this case.
6. In similar topic, what happen in desorption cases in breakthrough, I mean when the fixed bed is undergoing regeneration, have the authors quantified the effect of humidity in gas desorption cases and in cyclic performance?

Reviewer #2 (Remarks to the Author)

The authors reported a GeF_6^{2-} anion embedded MOF, ZNU-6, with optimized pore structure and environment for highly efficient C_2H_4 recovery from various $\text{C}_2\text{H}_2/\text{CO}_2/\text{C}_2\text{H}_4$ ternary mixtures. The material exhibited good recyclability and resistance toward moisture during breakthrough experiments. Ethylene (C_2H_4) purification from multi-component mixtures by physical adsorption is an important industrial challenging, and this work will be very interesting for the readers. The experiments were well performed and the manuscript has been written well. I agree that this manuscript to be accepted for Nature Communications after some minor revision. The following are the suggestions for the authors to improve the manuscript.

1. Figure 1 Caption, (f) is inconsistent with other numbering.
2. Supplementary Figure 10-11, the authors should provide the pressure of these IAST

selectivities.

3. For the in-situ gas-loaded MOF structure, the authors should add a figure to describe the C₂H₂ gas molecule cluster and the interactions between the molecules, either in main text or SI. A figure illustrating the interaction between adsorbed water and gas molecules should also be added.

Reviewer #3 (Remarks to the Author)

Thanks to its potential to introduce energy-efficient, single-step ethylene purification approaches, this manuscript by Jiang et al. is of high topical relevance to gas purifications, and MOFs for separations. Considering high importance of the new findings, and their general relevance in controlling the pore electrostatics (F···C=O interactions) driven gas separation/purification properties, I support publication subject to necessary revisions as follows:

1. Molecular formulae are missing including that of the new as-synthesized ZNU-6 (although I find this formula in Table S7, should be clearly noted in the manuscript too). All molecular formulae for the gas-loaded phases should be clearly written in the main article, if needed, using a table with analysis of sorbate-sorbent interactions (distances)
2. The authors should not present the adsorption data with respect to molecules per GeF₆²⁻ anion, this makes the data skewed in favour of this adsorbent, ZNU-6. Also, the units for adsorption uptakes and pressure need to be consistent. Right now, with mixed use of cm³/g, mmol/g, mol/mol and mmol/g, these are all mixed up; same mix-up is observed in the units used for pressure, bar and kPa. Collectively, all these mistakes come together in the manuscript Figure 2. I strongly recommend replotting the isotherm-based uptakes in one consistent pair of units: mmol/g and bar. This not only makes this manuscript coherent but also helps the whole community working in this area with regard to comparing performance parameters across different adsorbents.
3. Page 2, lines 33-34, the authors need to discuss “Presently, multi-step purification process is adopted for purification of C₂H₄ from C₂H₄/C₂H₂/CO₂ mixtures.” This needs to take cognisance of the recent reports on single-step C₂H₄ purification from ternary (1:1:1 for C₂H₂/C₂H₄/C₂H₆) and quaternary (1:1:1:1 for C₂H₂/C₂H₄/C₂H₆/CO₂) gas mixtures. Some examples include: Cao, JW. et al., Nat Commun. 2021, 12, 6507. Xu, Z. et al., Nat Commun. 2020, 11, 3163.
4. I don't quite agree with the authors use of the word “static” in the heading of Figure 2. Should just be written as “The sorption performance.” In another instance, the authors use this phrase “static adsorption” (page 11, line 202), which I object to.
5. It is stated that the C₂H₄ productivity is 309 mL/g, again this unit is far from the standard unit typically used in other literature reports in this area, i.e., mol kg⁻¹ h⁻¹
6. Importantly, the authors have calculated productivity by integrating the effluent flow rate of C₂H₄ (cm³/min). However, all the breakthrough curves (in Fig. 5) are showed in with the effluent concentration (C/C₀). It is obvious that integrating the concentration curve cannot give the amount, although a few references adopt this wrong method. Therefore, it is necessary to show the details for direct measurement and/or indirect calculation of the effluent flow rate and concentration in this study.

See Shen, J. et al., Nat Commun. 2020, 11, 6259 as a reference.

7. How common / rare is the 1D topology among the anion-pillared MOF library? It would greatly help the article/future readers, if the authors can address this by a thorough literature-based contextualisation of the structural topology of ZNU-6.

8. Section 7 in the supplementary information has a header "Breakthrough simulations and experiments", the following figure captions for Supplementary Figures 28-43 are all including "Experimental" in their figure captions, nonetheless. I wonder where the simulation-derived breakthrough data is?

Overall, an interesting idea executed by the authors that should advance this area in the near future. I will be glad to look at a suitably revised article, when ready.

Reviewer #4 (Remarks to the Author)

This manuscript by Jiang et al, report a metal organic framework (ZNU-6) for the application of simultaneous removal of C₂H₂ and CO₂ from C₂H₄ stream. This area is of important practical applications and has been extensively investigated in the past few years. The overall quality of this manuscript is high with detailed investigation of the structure characterisation of the MOF, study of its adsorption properties using isotherm and breakthrough experiments, and thorough investigation of the host-guest interactions. Below are some comments and questions from me, and I would recommend the publication of this manuscript after the authors fully address them:

1. I think it is necessary that the authors reference some highly relevant publications in the introduction, e.g. Hexafluorogermanate (GeFSIX) Anion-Functionalized Hybrid Ultramicroporous Materials for Efficiently Trapping Acetylene from Ethylene, Ind. Eng. Chem. Res. 2018, 57, 21, 7266–7274.

2. The cif. File for CO₂ loaded ZNU-6 shows an O-C-O bond angle of CO₂ being 157° instead of the theoretical 180°, could the authors please explain this discrepancy.

3. Figure 2f, the Q_{st} for CO₂, why it showed a sudden drop at 2 mmol/g coverage?

4. From the isotherms for CO₂ and C₂H₄ (Figure 2b, 2d), the saturation uptake for CO₂ and C₂H₄ are very close; and the kinetic data (supplementary figure 25) of CO₂ and C₂H₄ are also similar with C₂H₄ being slightly faster in adsorption. For Q_{st}, CO₂ started higher than C₂H₄ then fall to be lower than C₂H₄. These three parameters (uptake, kinetic, Q_{st}), all seem indicating the interaction between the gas molecules and the framework is very similar. In this case, how do the authors rationalise the observed separation in breakthrough experiments? what do the authors think is the really reason that ZNU-6 can retain CO₂ from mixtures containing mainly C₂H₄ and small percentage of CO₂.

Comments from Reviewer 1:

Overall comment. In this work authors reported GeF_6^{2-} anion incorporated MOF named ZNU-6. The MOF has optimum pore structure and environment for one step ethylene purification and also high selective for carbon dioxide separation as well. This research area is very promising especially today when we need to find better alternative for energy storage and separation. The authors investigated the MOF in different conditions and proved its viability and effectiveness. The chemical approach of this work is strong, and the results are clearly visible from the experiments

However here are some of my questions and concerns

Comment 1. *The topic of ethylene purification/ separation using MOF is not very new, even incorporation of anions in MOFs has been reported by the authors group recently (Wang et al, Nano Research 2022). It would be great if authors can describe the novelty of this research as suitable of Nature common.*

Author response: Thank you for your comment. The topic of using MOFs for ethylene purification is not very new indeed. Presently, ethylene purification from $\text{C}_2\text{H}_2/\text{C}_2\text{H}_4$, $\text{C}_2\text{H}_2/\text{C}_2\text{H}_4$, $\text{C}_2\text{H}_2/\text{C}_2\text{H}_4/\text{C}_2\text{H}_6$ has been realized by a plenty of MOFs materials. Compared with those, the investigation on the single-step C_2H_4 purification from $\text{C}_2\text{H}_2/\text{CO}_2/\text{C}_2\text{H}_4$ has been investigated much less (no more than 10 materials). The reference mentioned in comment 1 (Wang et al, Nano Research 2022) is for C_2H_2 purification from other mixtures, different from this work. So far, TIFSIX-17-Ni, NTU-65 and NTU-67 are the optimal materials. However, each material has significant drawback. The capacity of C_2H_2 (3.30 mmol/g) and CO_2 (2.20 mmol/g) is relatively low in TIFSIX-17-Ni due to the over-contracted channel. NTU-65 capture C_2H_2 and CO_2 by tuning the gate opening, but the applied temperature must be at 263 K because lower temperatures lead to the adsorption of all the gases while higher temperatures cause the exclusion of CO_2 . NTU-67 displays modest C_2H_2 (3.29 mmol/g) and CO_2 (2.04 mmol/g) capacity as well as reduced $\text{C}_2\text{H}_2/\text{C}_2\text{H}_4$ and $\text{CO}_2/\text{C}_2\text{H}_4$ selectivity compared to TIFSIX-17-Ni. Besides, the separation performance of NTU-67 is deteriorated under

humid conditions. Therefore, the trade-off between adsorption capacity and selectivity in the C₂H₄ purification from C₂H₂/CO₂/C₂H₄ is still challenging to overcome by existing porous materials. ZNU-6 reported in this work is a novel MOF with both narrow interlaced channels, large pores and abundant functional sites for record C₂H₂/CO₂/C₂H₄ separation. MOFs with such kind of optimal pore chemistry and pore size/shape are rare. Besides, such high stability in humid air and water is difficult to realize in anion pillared MOFs due to the weak Cu-N coordination bonds. Notably, our study include in-situ single crystal structure analysis, in-situ IR spectra analysis and DFT calculation. All the results are consistent and lead to the same conclusion. There has never been a report that provides such detailed mechanism study in C₂H₂/CO₂/C₂H₄ separation. Therefore, there are several novel points in this research and all of these can be found in the main text.

***Comment 2.** Its seems that the customized pore environment helps to achieve higher selectivity however DFT studies, and in-situ X ray have been performed to corroborate the statement. Have the authors performed any in-situ IR to verify the conclusions? This evidence will help strengthen the case*

Author response: Thank you for your suggestion, the in-situ IR spectroscopy was conducted on gas-loaded and activated samples, and the results have added into supplementary materials. New and obvious stretching bands that belong to C₂H₂ and CO₂ are observed in the C₂H₂ and CO₂ dosed single crystals. The $\nu_{\text{as}}(\text{C}_2\text{H}_2)$ and $\nu(\text{C}\equiv\text{C})$ stretching band of adsorbed C₂H₂ down-shifted to 3160 and 1720 cm⁻¹ respectively with reference to the gas-phase value at 3287 and 2500-1900 cm⁻¹, indicating the existence of guest-host interactions. Similarity, $\nu(\text{CO}_2)$ band also undergoes a downward shift from gas-phase value 2349 cm⁻¹ to 2335 cm⁻¹, showing the interactions between CO₂ and framework. In contrast, the stretching band of C₂H₄ is not obvious in C₂H₄@ZNU-6.

Modification:

Supplementary: Page 7 Figure 5

Supplementary Figure 5. In-situ IR spectra for **a.** activated ZNU-6 (black), - C_2H_2 @ZNU-6 (blue) and re-activated ZNU-6 (purple); **b.** activated ZNU-6 (black) and CO_2 @ZNU-6 (green); **c.** activated ZNU-6 (black) and C_2H_4 @ZNU-6 (red).

All the IR spectroscopic data are recorded in a Nicolet iS5 ATR-FTIR spectrometer.

The samples of gas-loaded crystals were prepared by the method described in **Preparation of gas loaded ZNU-6** in manuscript.

As shown in the Supplementary Figure 5, new and obvious stretching bands that belong to C_2H_2 and CO_2 are observed in the C_2H_2 and CO_2 dosed single crystals. The $\nu_{as}(C_2H_2)$ and $\nu(C\equiv C)$ stretching band of adsorbed C_2H_2 down-shifted to 3160 and 1720 cm^{-1} respectively with reference to the gas-phase value at 3287 and 2500-1900 cm^{-1} , indicating the existence of guest-host interactions. Similarity, $\nu(CO_2)$ band also undergoes a downward shift from gas-phase value 2349 cm^{-1} to 2335 cm^{-1} , showing the interactions between CO_2 and framework. In contrast, the stretching band of C_2H_4 is not obvious in C_2H_4 @ZNU-6.

***Comment 3.** Heat of adsorption values and isotherm dependency with temp indicates the physisorption nature of guest /host chemistry. However, the additional interaction with anions and even TPA can provide some chemisorption in CO₂. Have the authors performed any Temp programed desorption studies in terms of gases of choice to see the binding behavior?*

Author response: Thank you for your suggestion. Firstly, there is no hysteresis in the desorption curves, indicating that the interactions between gas molecules and framework are not too strong. Secondly, we have performed the desorption at room temperature. The reactivation condition between the 5th cycle and 6th cycle adsorption experiment is under dynamic vacuum at room temperature for three hours, as shown in Supplementary Fig. 26, the uptake of 6th cycle is similar with that of 5th cycle. Such mild regeneration condition shows that the interactions between CO₂ and ZNU-6 are relatively weak. Thus, it belongs to physisorption rather than chemisorption. Finally, as shown in in-situ crystals, the binding sites of CO₂ are close to GeF₆²⁻ anion in either large cage or small interlaced channel, this shows that the interactions between CO₂ and GeF₆²⁻ anion are stronger than those between CO₂ and TPA. Besides, due to the steric hinderance of pyridine rings, the CO₂ molecules are difficult to approach TPA ligands. In summary, we believe that the whole CO₂ adsorption behavior belongs to physisorption and the strongest interaction is between CO₂ and GeF₆²⁻ anion.

***Comment 4.** What is the relative humidity used in this study?*

Author response: The relative humidity used in the breakthrough experiments is 60%.

***Comment 5.** We can see ZNU-6 is strong water adsorbent however in humid conditions shows negligible deterioration of separation performance. However, the reasoning and data is not very clear, has the authors tried to quantify the effect of humidity in this case.*

Author response: We have quantified the effect of humidity on the breakthrough by calculating the C₂H₄ productivity. As shown in Table S9 in supplementary, the C₂H₄ productivity is 13.81 and 13.79 mol/kg respectively under dry and humid conditions. The productivity decrease is only 0.14%, in the range of measurement error . The reasons for the negligible deterioration are mainly attributed to the slow diffusion of water molecules in the framework. As shown in supplementary Fig 28 and 48, the H₂O adsorption is very slow. In real breakthrough conditions, less than 30% of saturated H₂O amount is adsorbed within 10.6 h while the gas mixture breakthrough experiments at 298 K are all finished within 200 min. The quite slow diffusion may be resulted from the small hydrophobic widows between large cage and interlaced channel. On the other hands, our in-situ single crystal structure have showed the water can be co-adsorbed in ZNU-6 without the reduction of gas loading. In the original main text (line 226-230), we have showed the reasons: “Although many water molecules can be adsorbed in ZNU-6, as described in in-situ crystals and water adsorption isotherms (Supplementary Fig. 27), the presence of humid has negligible influence on the separation performance (Fig. 5f). This is probably due to the co-adsorption of water and target gases as well as the fast C₂H₂/CO₂/C₂H₄ diffusion kinetics (Supplementary Fig. 29-31).”

Modifications:

Supplementary: Page 41 Figure 48

Supplementary Figure 48. H₂O uptake in breakthrough experiments (N₂, RH=100%).
Flow rate: 5 mL/min.

Supplementary: Page 42 Table S9

Supplementary Table S9. Experimental dynamic C₂H₄ productivity and captured C₂H₂/CO₂ amount for ZNU-6 from different gas ratios and under different conditions.

Conditions	Experimental C₂H₄ productivity (mol/kg)
C ₂ H ₂ -CO ₂ -C ₂ H ₄ (1-9-90) 298 K (dry)	13.81
C ₂ H ₂ -CO ₂ -C ₂ H ₄ (1-9-90) 298 K (humid)	13.79

Comment 6. In similar topic, what happen in desorption cases in breakthrough, I mean when the fixed bed is undergoing regeneration, have the authors quantified the effect of humidity in gas desorption cases and in cyclic performance?

Author response: Thank you for your comment. We have conducted the desorption experiments respectively after the breakthrough experiments under dry and humid

conditions, the results are shown in the graph below. It is obviously that the influence of moisture is negligible on the process of the regeneration. This is also reflected by the overlapping (adsorption) breakthrough curves (Fig 5f).

Comments from Reviewer 2:

Overall comment. The authors reported a GeF_6^{2-} anion embedded MOF, ZNU-6, with optimized pore structure and environment for highly efficient C_2H_4 recovery from various $\text{C}_2\text{H}_2/\text{CO}_2/\text{C}_2\text{H}_4$ ternary mixtures. The material exhibited good recyclability and resistance toward moisture during breakthrough experiments. Ethylene (C_2H_4) purification from multi-component mixtures by physical adsorption is an important industrial challenging, and this work will be very interesting for the readers. The experiments were well performed and the manuscript has been written well. I agree that this manuscript to be accepted for Nature Communications after some minor revision. The following are the suggestions for the authors to improve the manuscript.

Comment 1. Figure 1 Caption, (f) is inconsistent with other numbering.

Author response: Thanks for your reminder. We have corrected the Figure 1 Caption

Modification:

Manuscript: Page 5 Fig. 1

Fig. 1: a-c Exquisite control of pore size/shape and pore chemistry in ZNU-6 from pillared (3,4)-connected pto network to GeF_6^{2-} embedded itih-d topology framework; **d** Overview of ZNU-6 structure with cage-like pores and interlaced channels. **e** Structure and size of the cage-like pore. **f** Structure and size of the interlaced channel connecting four cages.

*Comment 2.*Supplementary Figure 10-11, the authors should provide the pressure of these IAST selectivities.

Author response: Thanks for your suggestion. We have added the pressure to the Supplementary Figure 13, 14.

Modification:

Supplementary: Page 14 Figure 13, 14

Supplementary Figure 13. IAST selectivity of ZNU-6 towards gas mixtures of C₂H₂/C₂H₄ with different ratios at 298 K and 1 bar.

Supplementary Figure 14. IAST selectivity of ZNU-6 towards gas mixtures of

CO₂/C₂H₄ with different ratios at 298 K and 1 bar.

Comment 3. For the in-situ gas-loaded MOF structure, the authors should add a figure to describe the C₂H₂ gas molecule cluster and the interactions between the molecules, either in main text or SI. A figure illustrating the interaction between adsorbed water and gas molecules should also be added.

Author response: Thanks for your suggestion. We have added the figures and the description to SI.

Modifications:

Supplementary: Page 6 Figure 3, 4

Supplementary Figure 3. The adsorption configuration of C₂H₂ molecules inside the narrow channel (site I) of ZNU-6 with the formation of rare C₂H₂ clusters. The C-H interaction and π...π packing distance is highlighted.

There are two kinds of interactions between C₂H₂ molecules in the site I. One is the C...H interactions, whose distances are between 2.2 and 2.6 Å, and the other is π...π interactions between C≡C bonds, which are all in the distance of 2.4 Å.

Supplementary Figure 4. Single crystals structure of gas loaded ZNU-6. a. $\text{CuGeF}_6\text{C}_{20}\text{H}_{16}\text{N}_{5.33}(\text{CO}_2)_3(\text{H}_2\text{O})_{3.274}$. b. $\text{CuGeF}_6\text{C}_{20}\text{H}_{16}\text{N}_{5.33}(\text{C}_2\text{H}_4)_{2.178}(\text{O})_{1.137}$. c. $\text{CuGeF}_6\text{C}_{20}\text{H}_{16}\text{N}_{5.33}(\text{C}_2\text{H}_2)_{4.296}(\text{O})_{0.517}$.

Due to the serious disorder of H atoms of H_2O molecules, we haven't solve the H atoms in C_2H_2 and C_2H_4 loaded ZNU-6. In CO_2 loaded crystal, besides 18 CO_2 molecules, there are 19.644 water molecules in each unit cell (sum formula $\text{Cu}_6\text{Ge}_6\text{F}_{36}\text{C}_{120}\text{H}_{96}\text{N}_{32}$). As to C_2H_4 loaded crystals, there are 13.068 C_2H_4 molecules and 6.822 H_2O molecules in an unit cell. In the C_2H_2 loaded crystals, the number of H_2O (3.102) is much lower than that of CO_2 or C_2H_4 loaded crystals. These H_2O vapor molecules don't occupy the adsorption site of targeted gas molecules. Instead, some weak interactions between H_2O and targeted gas molecules were observed. As shown above, The distances of H (H_2O) and O (CO_2) are 2.3-3.2 Å, those of O (H_2O) and H (C_2H_4) are 2.2 and 2.9 Å, and those of O (H_2O) and H (C_2H_2) are 3.2 Å.

Comments from Reviewer 3:

Thanks to its potential to introduce energy-efficient, single-step ethylene purification approaches, this manuscript by Jiang et al. is of high topical relevance to gas purifications, and MOFs for separations. Considering high importance of the new findings, and their general relevance in controlling the pore electrostatics ($\text{F}\cdots\text{C}=\text{O}$ interactions) driven gas separation/purification properties, I support publication subject to necessary revisions as follows:

Comment 1. *Molecular formulae are missing including that of the new as-synthesized*

ZNU-6 (although I find this formula in Table S7, should be clearly noted in the manuscript too). All molecular formulae for the gas-loaded phases should be clearly written in the main article, if needed, using a table with analysis of sorbate-sorbent interactions (distances).

Author response: Thank you for your valuable suggestion, we have added the formulae to the table and the corresponding places in the manuscript.

Modifications:

Manuscript: Page 4 Line 78-79

X-ray crystal analysis revealed that ZNU-6 ($\text{Cu}_6\text{Ge}_6\text{F}_{36}\text{C}_{120}\text{H}_{96}\text{N}_{32}$) crystallizes in a three-dimensional (3D) framework in the cubic Pm-3n space group.

Manuscript: Page 11 **Fig. 4**.

Fig. 4: Single crystal structure of gas-loaded ZNU-6. **a** C_2H_2 @ ZNU-6 [$\text{Cu}_6\text{Ge}_6\text{F}_{36}\text{C}_{120}\text{H}_{96}\text{N}_{32} (\text{C}_2\text{H}_2)_{25.78}$]; **b** CO_2 @ ZNU-6 [$\text{Cu}_6\text{Ge}_6\text{F}_{36}\text{C}_{120}\text{H}_{96}\text{N}_{32} (\text{CO}_2)_{18}$]; **c** C_2H_4 @ ZNU-6 [$\text{Cu}_6\text{Ge}_6\text{F}_{36}\text{C}_{120}\text{H}_{96}\text{N}_{32} (\text{C}_2\text{H}_4)_{13.07}$]; **d** Table of the distances of the host-guest interactions.

Comment 2. The authors should not present the adsorption data with respect to

molecules per GeF_6^{2-} anion, this makes the data skewed in favour of this adsorbent, ZNU-6. Also, the units for adsorption uptakes and pressure need to be consistent. Right now, with mixed use of cm^3/g , mmol/g , mol/mol and mmol/g , these are all mixed up; same mix-up is observed in the units used for pressure, bar and kPa. Collectively, all these mistakes come together in the manuscript Figure 2. I strongly recommend replotting the isotherm-based uptakes in one consistent pair of units: mmol/g and bar. This not only makes this manuscript coherent but also helps the whole community working in this area with regard to comparing performance parameters across different adsorbents.

Author response: Thanks for your suggestion, to make the manuscript coherent and also helps the whole community working in this area with regard to comparing performance parameters across different adsorbents. we have unified the pressure units to bar, and the uptake units to mmol/g in the manuscript, especially in the Fig. 2. The Fig. 2c that was the adsorption data with respect to molecules per anion before has been replaced by the adsorption data in the units of mmol/g as suggested and the origin Fig.2c has been moved to supplementary. However, we want to make an interpretation why we chose different units to present the adsorption data at first. Different units have different significance, mmol/g (or STP cm^3/g) is the basic uptake unit, the data in this units can be straightly obtained from the adsorption equilibrium measurements. While the density of single crystal is identified, the uptake data in cm^3/g can be converted to that in cm^3/cm^3 to evaluate the uptake of the adsorbent at a certain volume which is more useful for industrial application. To evaluate the influence of anions on the adsorption accurately, and to eliminate the effect brought by density and molecular mass simultaneously, mol/mol was chosen to be the uptake unit. The uptake data in mol/mol represents the number of the gas molecules those are adsorbed by per anion.

Modifications:

Manuscript: Page 2 Line 17

ZNU-6 exhibits significantly high C₂H₂ (1.53 mmol/g) and CO₂ (1.46 mmol/g) capacity at 0.01 bar.

Manuscript: Page 3 Line 47

However, the capacity of C₂H₂ (3.30 mmol/g) and CO₂ (2.20 mmol/g) is relatively low due to the over-contracted channel.

Manuscript: Page 3 Line 50-53

similar C₂H₂ (3.29 mmol/g) and CO₂ (2.04 mmol/g) capacity, but the C₂H₂/C₂H₄ and CO₂/C₂H₄ selectivity is greatly reduced as the C₂H₄ capacity (1.41 mmol/g) is relatively high.

Manuscript: Page 3 Line 58-61

Static gas adsorption isotherms showed that ZNU-6 takes up 1.53/8.06 mmol/g of C₂H₂ and 1.46/4.76 mmol/g of CO₂ at 0.01 and 1.0 bar (298 K), respectively. The calculated IAST selectivities for C₂H₂/C₂H₄ (1/99) and CO₂/C₂H₄ (1/99) are 43.8-14.3 and 52.6-7.8 (0.0001-1.0 bar), respectively.

Manuscript: Page 6 Line 104-106

At 1.0 bar, the C₂H₂ and CO₂ uptakes are 8.06 and 4.76 mmol/g (Fig 2c), higher than those of most APMOFs. The capacities are equal to 4.68 and 2.77 gas molecules per GeF₆²⁻ anion.

Manuscript: Page 6 Line 112-120

Notably, the uptakes of C₂H₂ and CO₂ on ZNU-6 at 0.01 bar are as high as 1.53 and 1.46 mmol/g, superior to those of all the porous materials in the context of ternary C₂H₂/CO₂/C₂H₄ separation, such as TIFSIX-17-Ni (1.38/0.32 mmol/g),³⁶ SIFSIX-17-Ni (0.91/0.20 mmol/g),³⁶ NTU-67 (0.47/0.65 mmol/g)³⁸ and TpPa-NO₂ (0.17/0.03 mmol/g)³⁹. At 0.1 bar, the capacities of C₂H₂ and CO₂ reach up to 4.64 and 2.21 mmol/g (Fig. 2b), even higher than the uptakes of many porous materials at 1 bar and 298 K, for example, TIFSIX-17-Ni (3.30/2.20 mmol/g).³⁶ In the meantime, the

C_2H_4 uptakes on ZNU-6 at 0.01 and 0.1 bar are only 0.15 and 1.07 mmol/g, much lower than those of C_2H_2 and CO_2 under the same conditions. The C_2H_2 , CO_2 and C_2H_4 adsorption isotherms were further collected at 278 and 308 K (Fig. 2d). The adsorption capacities of C_2H_2 and CO_2 at 1 bar increase to 8.74 and 6.26 mmol/g at 278 K.

Manuscript: Page 7 Fig.2

Fig. 2: The sorption performance. **a** N_2 adsorption and desorption isotherms for ZNU-6 and the calculated pore size distribution. **b** C_2H_2 , CO_2 and C_2H_4 adsorption isotherms of ZNU-6 at 298 K. **c** Comparison of the saturated C_2H_2 and CO_2 uptake (1 bar, 298 K) among anion pillared MOFs. **d** C_2H_2 , CO_2 and C_2H_4 adsorption isotherms of ZNU-6 at 278/308 K. **e** C_2H_2/C_2H_4 and CO_2/C_2H_4 IAST selectivity of ZNU-6 at 298 K. **f** Q_{st} of C_2H_2 , CO_2 and C_2H_4 in ZNU-6.

Comment 3. Page 2, lines 33-34, the authors need to discuss “Presently, multi-step purification process is adopted for purification of C_2H_4 from $C_2H_4/C_2H_2/CO_2$ mixtures.” This needs to take cognisance of the recent reports on single-step C_2H_4 purification from ternary (1:1:1 for $C_2H_2/C_2H_4/C_2H_6$) and quaternary (1:1:1:1 for $C_2H_2/C_2H_4/C_2H_6/CO_2$) gas mixtures. Some examples include: Cao, JW. et al., Nat

Commun. 2021, 12, 6507. Xu, Z. et al., *Nat Commun.* 2020, 11, 3163.

Author response: Thanks for your suggestion. However, we have to make a clarity: lines 33-34 in page 2 shows the industrial situation instead of the current status of C₂H₄ purification by physical adsorption. To avoid misunderstanding, we have modified the manuscript according to your advice, and we have added the references to the corresponding places in page 3 line 41-42.

Modifications:

Manuscript: Page 2 Line 33-34

Presently, multi-step purification process is adopted for purification of C₂H₄ from C₂H₄/C₂H₂/CO₂ mixtures in industry.

Manuscript: Page 3 Line 41-42

Besides, single-step purification of C₂H₄ from ternary C₂H₂/C₂H₄/C₂H₆^{33, 34} or quaternary C₂H₂/C₂H₄/C₂H₆/CO₂³⁵ mixtures has also been realized by several porous materials.

Manuscript: Page 22

33. Xu, Z. et al. A robust Th-azole framework for highly efficient purification of C₂H₄ from a C₂H₄/C₂H₂/C₂H₆ mixture. *Nat. Commun.* **11**, 3163 (2020).

34. Gu, X.-W. et al. Immobilization of Lewis Basic Sites into a Stable Ethane-Selective MOF Enabling One-Step Separation of Ethylene from a Ternary Mixture. *J. Am. Chem. Soc.* **144**, 2614-2623 (2022).

35. Cao, J. -W. One-step ethylene production from a four-component gas mixture by a single physisorbent. *Nat. Commun.* **12**, 6507 (2021).

Comment 4. I don't quite agree with the authors use of the word "static" in the heading of Figure 2. Should just be written as "The sorption performance." In another instance, the authors use this phrase "static adsorption" (page 11, line 202),

which I object to.

Author response: Thank you for your suggestion. We have modified the Figure 2 caption and the sentence in the manuscript.

Modifications:

Manuscript: Page 7-8 Fig.2

Fig. 2: The sorption performance.

Manuscript: Page 11, line 206-207

Motivated by the high adsorption capacity and selectivity in single-component adsorption as well as the in-situ single crystal structure analysis, breakthrough experiments were conducted for C₂H₂/C₂H₄, CO₂/C₂H₄ and C₂H₂/CO₂/C₂H₄ mixtures.

Comment 5. It is stated that the C₂H₄ productivity is 309 mL/g, again this unit is far from the standard unit typically used in other literature reports in this area, i.e., mol kg⁻¹ h⁻¹

Author response: Thank you for the reminder. However, the flow rate play a key role on the time of breakthrough experiment, as shown in Supplementary Fig. 34, 35, it is obvious that although the C₂H₂ captured amount is similar, the break time of C₂H₂ is evidently different under different flow rate. Therefore, it is unable to compare the separation performance with other materials under different flow rate if choosing mol kg⁻¹ h⁻¹ as the unit. Thus, we modified the units of productivity and captured amount to mol/kg, which is used quite often in literature.

Modifications:

Manuscript: Page 11-12, Line 211-220

For 1/1/98 C₂H₂/CO₂/C₂H₄ mixtures, C₂H₂ and CO₂ broke out simultaneously and 64.42 mol/kg of polymer grade C₂H₄ is produced by single adsorption process (Fig. 5b). The productivity is improved to 80.89 mol/kg when decreasing the temperature to

283 K (Supplementary Fig. 42). The CO₂ breakthrough time becomes shortened with the increase of CO₂ ratio, which is 72 and 52 mins for 1/5/94 (Fig. 5c) and 1/9/90 (Fig. 5d) C₂H₂/CO₂/C₂H₄ mixtures. The polymer grade C₂H₄ productivity is 21.37 and 13.81 mmol/kg, respectively. As most reported C₂H₄ productivity from C₂H₂/CO₂/C₂H₄ mixtures are compared under 1/9/90, a comparison plot of the C₂H₄ productivity and dynamic C₂H₂ capacity from 1/9/90 C₂H₂/CO₂/C₂H₄ mixtures is presented in Fig. 5e. ZNU-6 displays the record high C₂H₄ productivity and second highest C₂H₂ dynamic capacity. The C₂H₄ productivity of ZNU-6 is >2.5 folds of the previous benchmark of NTU-67 (5.42 mol/kg).

Manuscript: Page 13, line 245

The C₂H₂/anion and CO₂/anion uptakes are the highest among all the anion pillared MOFs. 64.42, 21.37, 13.81 mol/kg polymer grade C₂H₄ can be produced from C₂H₂/CO₂/C₂H₄ (1/1/98, 1/5/94, 1/9/90) mixtures, all superior to the previous benchmarks.

Manuscript: Page 13, Fig. 5e

Supplementary: Page 42, Table S9

Supplementary Table S9. Experimental dynamic C₂H₄ productivity and captured C₂H₂/CO₂ amount for ZNU-6 from different gas ratios and under different conditions.

Conditions	Experimental C₂H₄ productivity (mol/kg)	Experimental C₂H₂ captured amount (mol/kg)	Experimental CO₂ captured amount (mol/kg)
C ₂ H ₂ -CO ₂ -C ₂ H ₄ (1-1-98) 283 K	80.89	0.96	0.98
C ₂ H ₂ -CO ₂ -C ₂ H ₄ (1-1-98) 298 K	64.42	0.78	0.84
C ₂ H ₂ -CO ₂ -C ₂ H ₄ (1-1-98) 323 K	36.73	0.48	0.53
C ₂ H ₂ -CO ₂ -C ₂ H ₄ (1-5-94) 298 K	21.37	0.60	1.52
C ₂ H ₂ -CO ₂ -C ₂ H ₄ (1-9-90) 298 K	13.81	0.56	1.97
C ₂ H ₂ -CO ₂ -C ₂ H ₄ (5-5-90) 298 K	11.04	2.65	0.55
C ₂ H ₂ -CO ₂ -C ₂ H ₄ (1-9-90) 298 K (humid)	13.79	-	-

Comment 6. Importantly, the authors have calculated productivity by integrating the effluent flow rate of C₂H₄ (cm³/min). However, all the breakthrough curves (in Fig. 5) are showed in with the effluent concentration (C/C₀). It is obvious that integrating the concentration curve cannot give the amount, although a few references adopt this wrong method. Therefore, it is necessary to show the details for direct measurement and/or indirect calculation of the effluent flow rate and concentration in this study.

See Shen, J. et al., Nat Commun. 2020, 11, 6259 as a reference.

Author response: Thank you for your question. However, we need to clarify one point that we choose mL/g as the unit of x-axis instead of cm^3/min , the normalized flow volume is calculated by the formula "Flow volume (mL/g)= flowrate (mL/min) \times time (min) /sample weight (g)". The real flow rates and original figures of experiments with time (min) as x-axis had been presented in the supplementary. According to formula and the figure below, the flow rates have been considered in the calculation of C_2H_4 productivity. Therefore, the productivity can be straightly calculated out by the y-axis which represents effluent concentration (C/C_0).

$$\text{Flow volume (mL/g)} = \text{flowrate (mL/min)} \times \text{time (min)} / \text{sample weight (g)}$$

$$\text{C}_2\text{H}_4 \text{ productivity} = \text{S1 (min)} \times \text{flowrate (mL/min)} \times \text{time (min)} / \text{sample weight (g)} = \text{S2 (mL/g)}$$

Comment 7. How common / rare is the *ith-d* topology among the anion-pillared MOF library? It would greatly help the article/future readers, if the authors can address this by a thorough literature-based contextualisation of the structural topology of ZNU-6.

Author response: Thank you for your reminder. To help the article/future readers understand clearly, we will give an introduction. According to the previous research papers and reviews [Li, Xu. et al. *Coord. Chem. Rev.* **470**, 214714 (2022)], among the APMOFs, just two MOFs with *ith-d* topology have been reported previously, namely SIFSIX-Cu-TPA [Li, H. et al. *Angew. Chem. Int. Ed.* **60**, 7547-7552 (2021)], ZNU-2

[TIFSIX-Cu-TPA, Jiang, Y. et al. *Angew. Chem. Int. Ed.* **61**, e202200947 (2022)].

Comment 8. Section 7 in the supplementary information has a header "Breakthrough simulations and experiments", the following figure captions for Supplementary Figures 28-43 are all including "Experimental" in their figure captions, nonetheless. I wonder where the simulation-derived breakthrough data is?

Author response: Thank you for your kind reminder, we have corrected the header of section 7.

Modification:

Supplementary: Page 33

V Breakthrough experiments

Overall, an interesting idea executed by the authors that should advance this area in the near future. I will be glad to look at a suitably revised article, when ready.

Comments from Reviewer 4:

This manuscript by Jiang et al, report a metal organic framework (ZNU-6) for the application of simultaneous removal of C₂H₂ and CO₂ from C₂H₄ stream. This area is of important practical applications and has been extensively investigated in the past few years. The overall quality of this manuscript is high with detailed investigation of the structure characterisation of the MOF, study of its adsorption properties using isotherm and breakthrough experiments, and thorough investigation of the host-guest interactions. Below are some comments and questions from me, and I would recommend the publication of this manuscript after the authors fully address them:

Comment 1. *I think it is necessary that the authors reference some highly relevant publications in the introduction, e.g. Hexafluorogermanate (GeFSIX)*

Anion-Functionalized Hybrid Ultramicroporous Materials for Efficiently Trapping Acetylene from Ethylene, Ind. Eng. Chem. Res. 2018, 57, 21, 7266–7274.

Author response: Thank you for your suggestion, we have added some references to the introduction, such as ref. 25, 26.

Modifications:

Manuscript: Page 21

25. Zhang, Z.-Q. et al. Hexafluorogermanate (GeFSIX) Anion-Functionalized Hybrid Ultramicroporous Materials for Efficiently Trapping Acetylene from Ethylene. *Ind. Eng. Chem. Res.* **57**, 7266-7274 (2018).

26. Ke, Tian. et al. Molecular Sieving of C₂-C₃ Alkene from Alkyne with Tuned Threshold Pressure in Robust Layered Metal-Organic Frameworks. *Angew.Chem. Int. Ed.* **59**, 12725-12730 (2020).

Comment 2. The cif. File for CO₂ loaded ZNU-6 shows an O-C-O bond angle of CO₂ being 157° instead of the theoretical 180°, could the authors please explain this discrepancy.

Author response: Thank you for your valuable comment. The slight distortion of CO₂ is caused by the relatively strong interaction between F atoms and C atom of CO₂. Similar phenomena have been observed in other research papers [*Chem. Commun.* **41**, 5125-5127 (2008), *Chem* **5**, 950-963 (2019)].

CO₂ @ CPO-27-Ni [Chem. Commun. 41, 5125-5127 (2008)] The angle of C-O-C is 162°

CO₂ @ dptz-CuTiF₆ [Chem 5, 950-963 (2019)] The angle of C-O-C is 167°

Comment 3. Figure 2f, the Q_{st} for CO₂, why it showed a sudden drop at 2 mmol/g coverage?

Author response: Thank you for your comment. From in-situ crystal CO₂ @ ZNU-6, two binding sites can be observed, and the ratio of the number of CO₂ molecules adsorbed in the site I/II is 1:2. According to the DFT calculations, the binding energy in site I is stronger than that in site II, so CO₂ prefer to be adsorbed in site I. Therefore, at the first stage of the whole adsorption, CO₂ is adsorbed in site I until the uptake in site I reach to saturated point (≈ 1.72 mmol/g), at the following stage, CO₂ begin to be adsorbed in the site II, while the binding energy in site II is lower, so the whole Q_{st} begins to decrease. Because the ratio of CO₂ molecules adsorbed in site I/II is 1:2, the decrease of Q_{st} is sharp. Line 173-174 has described the phenomenon.

Comment 4. From the isotherms for CO₂ and C₂H₄ (Figure 2b, 2d), the saturation uptake for CO₂ and C₂H₄ are very close; and the kinetic data (supplementary figure 25) of CO₂ and C₂H₄ are also similar with C₂H₄ being slightly faster in adsorption. For Q_{st} , CO₂ started higher than C₂H₄ then fall to be lower than C₂H₄. These three parameters (uptake, kinetic, Q_{st}), all seem indicating the interaction between the gas

molecules and the framework is very similar. In this case, how do the authors rationalise the observed separation in breakthrough experiments? what do the authors think is the really reason that ZNU-6 can retain CO₂ from mixtures containing mainly C₂H₄ and small percentage of CO₂.

Author response: Thank you for your valuable comment. The near-zero loading Q_{st} is more important to distinguish the adsorption affinity. Thus, CO₂ and C₂H₂ are more favored to be adsorbed in ZNU-6. Besides, according to the DFT calculations, C₂H₂, CO₂ and C₂H₄ are all preferentially adsorbed in the site I, and according to the binding energy, the affinity sequence in site I is also C₂H₂~CO₂>C₂H₄. From the in-situ crystals, nearly all C₂H₄ molecules are adsorbed in the site I. So the Q_{st} of C₂H₄ calculated from single-component adsorption isotherms reflects the isosteric enthalpy of C₂H₄ in the interlaced channel. However, when the single-component C₂H₄ adsorption changes to competitive C₂H₂/CO₂/C₂H₄ mixture adsorption, the situation will become different. Because the affinity towards C₂H₂ and CO₂ in site I is higher than that towards C₂H₄, the C₂H₂ and CO₂ gases will occupy the interlaced channel (site I) and C₂H₄ adsorption sites will have to be changed to the large cage (site II). While in site II, the binding energy of C₂H₄ is much lower. In brief, it is mainly the much higher Q_{st} of CO₂ at near-zero loading that distinguishes CO₂ from C₂H₄.

REVIEWERS' COMMENTS

Reviewer #1 (Remarks to the Author):

The authors in this manuscript have very reasonably addressed all of my and other reviewers' comments and made/performed the necessary changes and experiments required for the quality of the manuscript. I agree with the technical soundness of this paper. I have no other concerns or reservations about this manuscript.

Reviewer #2 (Remarks to the Author):

The authors have done a good job of improving the manuscript by performing necessary experiments and analyses. And they have answered the all questions of the reviewers. I agree that this manuscript now can be published on Nature Communications.

Reviewer #3 (Remarks to the Author):

I will be glad to see a revised article published, below are my revision comments:

- Regarding the molecular formula, the authors write $\text{Cu}_6\text{Ge}_6\text{F}_{36}\text{C}_{120}\text{H}_{96}\text{N}_{32}$ as the formula. This is not the right way of writing formula for MOFs, the authors need to revise all these places having formula with something like $[\text{Cu}_n(\text{TPA})_x(\text{GeF}_6)_y]$, where, n , x and y are integers suites to the molecular formula of ZNU-6.
- Regarding the use of units mmol/g and bar, I am satisfied with the revisions done by the authors.
- Glad about the authors edits on including a few updated citations from 2020, 2021, and 2022.
- I appreciate the authors omitting the word "static" before "adsorption".
- I do not quite agree with the authors arguing that use of the more commonly used unit for productivity, $\text{mol kg}^{-1} \text{ h}^{-1}$ will neglect flow rate. The calculation of productivity with respect to this unit takes into account the flow rate used, and therefore, the authors are again advised to use the unit $\text{mol kg}^{-1} \text{ h}^{-1}$ for quantifying the C_2H_4 productivity.
- I am glad about the authors argument on the productivity calculations based on the plots presented in C/C_0 vs. time (min).
- The authors' response (in the revised introduction) on quantifying the rarity of 1D topology among anion-pillared family of MOFs is satisfactory.

Overall, I will be glad to look at a suitably revised article, when ready.

Reviewer #4 (Remarks to the Author):

The authors have fully addressed my comments, and I feel the manuscript is ready for publication.

Reviewer #3 (Remarks to the Author):

I will be glad to see a revised article published, below are my revision comments:

1. Regarding the molecular formula, the authors write $\text{Cu}_6\text{Ge}_6\text{F}_{36}\text{C}_{120}\text{H}_{96}\text{N}_{32}$ as the formula. This is not the right way of writing formula for MOFs, the authors need to revise all these places having formula with something like $[\text{Cu}_n(\text{TPA})_x(\text{GeF}_6)_y]$, where, n, x and y are integers suites to the molecular formula of ZNU-6.
2. Regarding the use of units mmol/g and bar, I am satisfied with the revisions done by the authors.
3. Glad about the authors edits on including a few updated citations from 2020, 2021, and 2022.
4. I appreciate the authors omitting the word “static” before “adsorption”.
5. I do not quite agree with the authors arguing that use of the more commonly used unit for productivity, mol kg⁻¹ h⁻¹ will neglect flow rate. The calculation of productivity with respect to this unit takes into account the flow rate used, and therefore, the authors are again advised to use the unit mol kg⁻¹ h⁻¹ for quantifying the C₂H₄ productivity.
6. I am glad about the authors argument on the productivity calculations based on the plots presented in C/C₀ vs. time (min).
- 7 The authors’ response (in the revised introduction) on quantifying the rarity of ith-d topology among anion-pillared family of MOFs is satisfactory.

Overall, I will be glad to look at a suitably revised article, when ready.

I will be glad to see a revised article published, below are my revision comments:

Comment 1 Regarding the molecular formula, the authors write $Cu_6Ge_6F_{36}C_{120}H_{96}N_{32}$ as the formula. This is not the right way of writing formula for MOFs, the authors need to revise all these places having formula with something like $[Cu_n(TPA)_x(GeF_6)_y]$, where, n, x and y are integers suites to the molecular formula of ZNU-6.

Author response: Thank you for your suggestion. We have modified the molecular formula in the manuscript and supplementary material.

Modification:

Manuscript: Page 4 Line 78-79

X-ray crystal analysis revealed that ZNU-6 $[Cu_6(GeF_6)_6(TPA)_8]$ crystallizes in a three-dimensional (3D) framework in the cubic Pm-3n space group.

Supplementary: Table S1.

Formula	$C_{20}H_{16}Cu$ $F_6GeN_{5.33}$ $Cu(GeF_6)(TPA)_{1.33}$	$C_{20}H_{16}CuGeF_6$ $N_{5.33} \cdot 4.296C_2H_2$ $Cu(GeF_6)(TPA)_{1.33}$ $(C_2H_2)_{4.296}$	$C_{20}H_{16}CuGeF_6$ $N_{5.33} \cdot 2.178C_2H_4$ $Cu(GeF_6)(TPA)_{1.33}$ $(C_2H_4)_{2.178}$	$C_{20}H_{16}CuGeF_6$ $N_{5.33} \cdot 3CO_2$ $Cu(GeF_6)(TPA)_{1.33}$ $(CO_2)_3$
---------	--	--	--	--

Comment 2 Regarding the use of units mmol/g and bar, I am satisfied with the revisions done by the authors.

Author response: Thank you for your positive comment.

Comment 3 Glad about the authors edits on including a few updated citations from 2020, 2021, and 2022.

Author response: Thank you for your positive comment.

Comment 4 I appreciate the authors omitting the word “static” before “adsorption”.

Author response: Thank you for your positive comment.

Comment 5 I do not quite agree with the authors arguing that use of the more commonly used unit for productivity, $mol\ kg^{-1}\ h^{-1}$ will neglect flow rate. The calculation of productivity with respect to this unit takes into account the flow rate used, and therefore, the authors are again advised to use the unit $mol\ kg^{-1}\ h^{-1}$ for quantifying the C_2H_4 productivity.

Author response: Thank you for your suggestion. We have added the productivity in the unit of mol kg⁻¹ h⁻¹ to the corresponding places. We have chose two time period, one is from 0 to **Time 1**, and the other is from 0 to **Time 2** as shown in Supplementary Table 10..

Modification:

Manuscript: Page 12 Line 220-222

C₂H₄ productivity with the unit of mol/kg/h is also calculated for comparison (Supplementary Table S10). ZNU-6 with the productivity of 15.93 mol/kg/h is still the best material.

Supplementary: Table 10/11

Supplementary Table S10. Experimental dynamic C₂H₄ productivity for different adsorbents.

	C ₂ H ₂ /CO ₂ /C ₂ H ₄ =1/9/90 (v/v/v) Flow rate: 5 mL/min					
	ZNU-6	NTU-67	Activated carbon	zeolite 5A	SIFSIX-2-Cu-i	SIFSIX-17-Ni
Mass (g)	0.58	1.20	0.98	1.74	0.52	0.82
Time 1 ^a (min)	52.00	43.59	21.40	13.18	12.61	12.24
Time 2 ^b (min)	196.00	84.38	34.19	45.89	170.08	83.82
Time 1 (min g ⁻¹)	89.56	36.17	21.84	7.58	24.20	14.96
Time 2 (min g ⁻¹)	337.58	70.03	34.89	26.40	326.44	102.42
Productivity per adsorption cycle (mol kg ⁻¹)	13.81	5.42	0.49	0.36	2.40	2.47
Productivity based on Time 1 (mol kg ⁻¹ h ⁻¹)	15.93	7.46	1.38	1.62	11.41	12.10
Productivity based on Time 2 (mol kg ⁻¹ h ⁻¹)	4.23	3.85	0.86	0.46	0.85	1.77
^a Time 1 is the time when the second gas can be detected after C ₂ H ₄ ;						
^b Time 2 is the time when C _A /C ₀ reaches 1.0 for all the gases.						

Supplementary Table S11. Experimental dynamic C_2H_4 productivity for ZNU-6 from different gas ratios and under different conditions.

	ZNU-6 Flow rate: 5 mL/min						
$C_2H_2/CO_2/C_2H_4$ (v/v/v)	1-1-98 283 K	1-1-98 298 K	1-1-98 323 K	1-5-94	1-9-90 dry	5-5-90	1-9-90 humid
Time 1 (min)	232.00	184.00	112.00	72.00	52.00	44.00	112.00
Time 2 (min)	284.00	248.00	180.00	192.00	196.00	180.00	180.00
Productivity per adsorption cycle (mol kg ⁻¹)	80.89	64.42	36.73	21.37	13.81	11.04	13.79
Productivity based on Time 1 (mol kg ⁻¹ h ⁻¹)	20.92	21.01	19.68	17.81	15.93	15.05	7.39
Productivity based on Time 2 (mol kg ⁻¹ h ⁻¹)	17.09	15.59	12.24	6.68	4.23	3.68	4.60

Comment 6 I am glad about the authors argument on the productivity calculations based on the plots presented in C/C_0 vs. time (min).

Author response: Thank you for your positive comment.

Comment 7 The authors' response (in the revised introduction) on quantifying the rarity of *ith-d* topology among anion-pillared family of MOFs is satisfactory.

Author response: Thank you for your positive comment.

Overall, I will be glad to look at a suitably revised article, when ready